# Mitigating Intra- and Inter-modal Forgetting in Continual Learning of Unified Multimodal Models

Xiwen Wei, Mustafa Munir, Radu Marculescu

Department of Electrical and Computer Engineering
The University of Texas at Austin
{xiwenwei, mmunir, radum}@utexas.edu

## Abstract

Unified Multimodal Generative Models (UMGMs) unify visual understanding and image generation within a single autoregressive framework. However, their ability to continually learn new tasks is severely hindered by catastrophic forgetting, both within a modality (intra-modal) and across modalities (inter-modal). While intra-modal forgetting has been studied in prior continual learning (CL) work, inter-modal forgetting remains largely unexplored. In this paper, we identify and empirically validate this phenomenon in UMGMs and provide a theoretical explanation rooted in gradient conflict between modalities. To address both intra- and inter-modal forgetting, we propose **Modality-Decoupled Experts (MoDE)**, a lightweight and scalable architecture that isolates modality-specific updates to mitigate the gradient conflict and leverages knowledge distillation to prevent catastrophic forgetting and preserve pre-trained capabilities. Unlike previous CL methods that remain modality-coupled and suffer from modality gradient conflict, MoDE explicitly decouples modalities to prevent interference. Experiments across diverse benchmarks demonstrate that MoDE significantly mitigates both inter- and intra-modal forgetting, outperforming prior CL baselines in unified multimodal generation settings. *Codes are publicly available: https://github.com/Christina200/MoDE-official.git.*

## 1 Introduction

Traditional multimodal models are typically divided into two categories: multimodal understanding (e.g. answering questions about images) and multimodal generation (e.g. generating images from text prompts) [1, 2]. Unified Multimodal Generative Models (UMGMs) aim to integrate these two tasks within a single framework. Recent advancements in UMGMs [3–8] have demonstrated strong performance on a wide range of tasks, such as visual question answering (VQA), image captioning, visual reasoning, classification, reading comprehension, and image generation. These models typically embed multiple input types into a shared representation space and leverage one single transformer backbone to model cross-modal interactions. Training generally follows a two-stage paradigm: an initial pretraining phase focused on text–image alignment, followed by fine-tuning for specific downstream tasks. During the second stage, instruction tuning (which fine-tunes the model on diverse, task-specific instructions paired with expected outputs) has emerged as a widely adopted strategy to better align model behavior with human intent [9, 10].

While UMGMs demonstrate impressive zero-shot performance on a wide range of unseen instructions, their effectiveness is inconsistent across all tasks, particularly when the relevant task data is absent from the pretraining corpus. Expanding the training dataset to include new task data can significantly improve performance on those tasks. However, given the continual emergence of novel multimodal

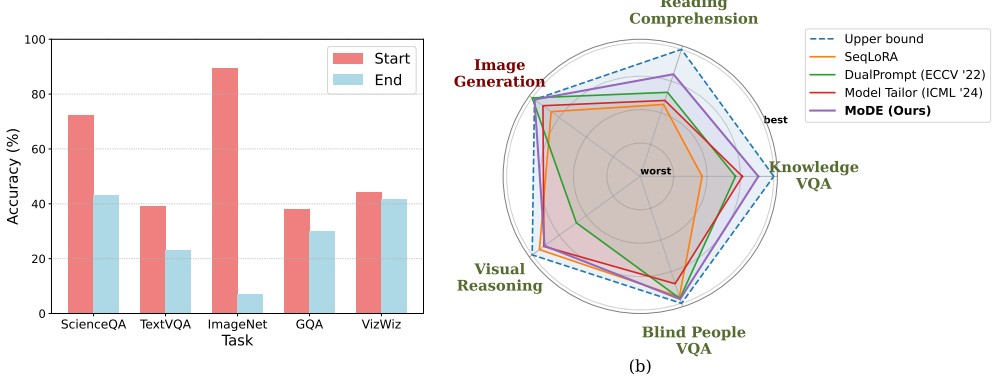

(b)

Figure 1: (a) illustrates catastrophic forgetting during naive sequential instruction tuning. Continual instruction tuning starts with the Chameleon [3] model on tasks ScienceQA→TextVQA→ImageNet→GQA→VizWiz. "Start" refers to the performance immediately after the model has been tuned on a task, while "End" denotes performance after completing all tasks. A larger gap between the two bars indicates more severe forgetting. (b) visualizes forgetting in both multimodal generation tasks (in **red**, representing inter-modal forgetting) and multimodal understanding tasks (in **green**, representing intra-modal forgetting). Our proposed MoDE mitigates both types of forgetting, preserving performance across modalities.

tasks, repeatedly merging new data and retraining the model is computationally expensive and impractical. This motivates the need for new approaches that can enable UMGMs to flexibly and efficiently adapt over time. This goal aligns with the principles of continual learning (CL), where models are designed to acquire capabilities incrementally, similar to how humans learn [11].

Existing studies in CL have shown that sequentially fine-tuned models suffer from catastrophic forgetting, a phenomenon where a model fine-tuned on new tasks tends to forget or overwrite the previously acquired knowledge [12]. More recently, several works have explored continual instruction tuning for multimodal large language models (MLLMs) [13–15]. However, these models are limited to language-only outputs and, as a result, prior research has focused exclusively on tasks such as VQA and other multimodal understanding benchmarks. In contrast, UMGMs are capable of both multimodal understanding and generation within a single backbone, enabling the production of *both* textual and visual outputs and thus representing a fundamentally different paradigm. This raises a new and largely unexplored challenge, whereby forgetting occurs not only *within* a single modality (intra-modal), but also *across* modalities (inter-modal) during continual instruction tuning of UMGMs.

Starting from these overarching ideas, we ask: *Can UMGMs continually acquire new capabilities through instruction tuning without suffering catastrophic forgetting, both within a modality (e.g., multimodal understanding) and across modalities (e.g., improving multimodal understanding while preserving visual generation)?*

As shown in Figure 1, UMGMs do experience catastrophic forgetting across a range of tasks, including visual reasoning, classification, VQA, and image generation. Also, while existing methods [16, 17] can mitigate either intra-modal or inter-modal forgetting, none of them can effectively address both issues simultaneously.

To address this fundamental challenge, we propose **Modality-Decoupled Experts (MoDE)**, a new lightweight architecture which incorporates a modality-aware sparse mixture of LoRA adapters for text (T-MoE) and a single LoRA adapter for images (V-Adapter). The intra-modal forgetting during continual multimodal understanding tasks is mitigated by the T-MoE's routing, which selectively activates the appropriate experts. The *inter-modal forgetting* in image generation is prevented through modality decoupling and knowledge distillation between the pre-trained model (teacher) and the newly added image LoRA (student). During continual instruction tuning, only the MoDE components are trained, while the pre-trained UMGM parameters remain frozen to preserve their robust cross-

modal alignment. Our results show that this design simultaneously mitigates both intra-modal and inter-modal catastrophic forgetting.

Our contributions are:

1. We identify and illustrate the unique challenge of inter-modal catastrophic forgetting, in addition to the more common intra-modal forgetting in autoregressive transformers that unify multimodal understanding and generation under continual instruction tuning. We further provide a theoretical analysis attributing inter-modal forgetting to *modality gradient conflict*, and formally prove that modality decoupling can mitigate this conflict both theoretically and experimentally.

2. We propose Modality-Decoupled Experts (MoDE), a new lightweight and scalable architecture that tackles both intra- and inter-modal forgetting by decoupling modality-specific updates (to reduce inter-modal gradient conflict) and applying knowledge distillation to preserve pre-trained image generation capabilities.

3. Through extensive experiments, we show that MoDE outperforms SOTA baselines (e.g. CL-MoE [18], Model Tailor [17]) when mitigating both intra- and inter catastrophic forgetting simultaneously. This opens up new possibilities for scalable continual learning in UMGMs.

The remainder of this paper is organized as follows: Section 2 reviews related work. Section 3 introduces inter-modal forgetting in UMGMs. Section 4 presents our proposed method, and Section 5 details the experimental results. Finally, Section 6 concludes our paper.

## 2 Related work

### 2.1 Unified multimodal generative models (UMGM)

Early efforts to unify visual understanding and generation [19–22] typically combine diffusion models with MLLMs, conditioning the diffusion process on LLM-generated embeddings. While effective for certain tasks, this "diffusion+MLLM" design lacks tight coupling between image generation and language modeling, resulting in suboptimal performance on instruction-based generation [23].

More recent approaches treat both understanding and generation as a unified next-token prediction problem [24–26, 3]. These models differ in their encoder and decoder designs, e.g. some [3, 6, 27] employ vision tokenizers such as VQGAN [28] to discretize images into token sequences, while others [29, 30] use semantic encoders like CLIP [31] or SigLIP [32] to represent images as sequences of continuous embeddings. Their decoders typically rely on either a VQVAE [33] or a diffusion-based generator [34, 22], but all maintain an autoregressive transformer backbone.

Our method is designed for transformer-based UMGMs and works with a broad class of models built on this autoregressive architecture, regardless of their choice of encoding or decoding mechanism.

### 2.2 Mixture of experts (MoE)

The Mixture of Experts architecture [35–38] combines specialized experts with a gating mechanism that efficiently allocates tokens. Several studies explore integrating MoE with LoRA [39] to facilitate scalable and efficient training of MLLMs and LLMs [40–45]. Recently, the scalability and modularity of MoE have attracted growing attention in continual learning [18, 46, 47]. For instance, Lifelong-MoE [48] introduces a strategy to train new experts while freezing the old ones. However, prior MoE-based CL methods share the same set of experts across modalities, which leads to modality interference during multimodal generation and exacerbates the inter-modal forgetting. Motivated by these limitations, we propose modality-decoupled MoE with LoRA, aiming at mitigating both inter-modal forgetting caused by modality interference and intra-modal forgetting within each modality.

### 2.3 Catastrophic Forgetting

Catastrophic forgetting remains a fundamental challenge in multimodal continual instruction tuning (MCIT), where a multimodal generative model is incrementally adapted to new datasets and task instructions without costly retraining from scratch. Existing methods to address catastrophic forgetting in MCIT typically fall into four categories: regularization-based [49], architecture-based [50, 14],

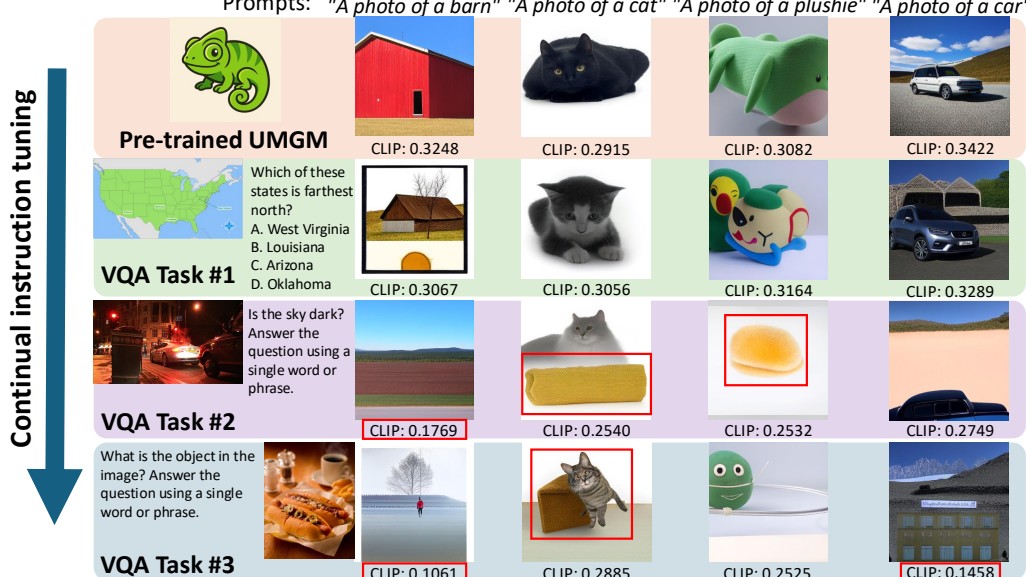

Figure 2: Inter-modal catastrophic forgetting during continual instruction tuning across three VQA tasks. The pre-trained UMGM serves as the upper bound for image generation quality. CLIP scores under each sample reflect text-image alignment in CLIP feature space. Red bounding boxes highlight regions with degraded image quality and low CLIP scores, indicating increasing misalignment between prompts and generated images. For instance, the image generated for the prompt "*A photo of a car*" depicts a building instead of a car (VQA task #3).

replay-based [15], and prompt-based approaches [51, 52]. For instance, EProj [11] applies task-similarity-based regularization to constrain model updates that are important for previous tasks. Fwd-Prompt [51] selects prompts using both textual and visual features to preserve earlier knowledge. LLaCA [50] employs exponential moving average (EMA) to merge the weights of old and new models, thereby maintaining both past task performance and pre-trained capabilities.

However, these approaches address only the *intra-modal catastrophic forgetting*, where old tasks, new tasks, and pre-trained capabilities all belong to the same output modality, typically text (e.g., in VQA tasks) [52–54]. While this assumption holds for traditional multimodal models with unimodal outputs, it is insufficient for UMGMs which generate multimodal outputs such as text and images.

## 3 Inter-modal Catastrophic Forgetting in UMGMs

Inter-modal catastrophic forgetting can manifest in several forms. For instance, prior work [55] examined the loss of text-only generation abilities in MLLMs when compared to their original LLM counterparts. Other studies have explored modality decoupling strategies in encoders and adapters of UMGMs [56, 57]. In this paper, we restrict our focus to multimodal tasks, specifically the forgetting that occurs in multimodal understanding and generation. In particular, we study visual generation forgetting that arises when tuning models on understanding tasks. Complementary results on understanding forgetting induced by tuning on image generation tasks are provided in Appendix F.

To investigate inter-modal catastrophic forgetting in UMGMs from the visual perspective, we conduct a preliminary experiment by sequentially tuning Chameleon [3] using LoRA on three VQA datasets [58–60], while monitoring image generation performance with prompt-based evaluation. As shown in Figure 2, we observe two key issues in image generation as tuning progresses: 1) The overall image quality degrades significantly, 2) The alignment between the generated image and the input prompt worsens over time.

These findings indicate that inter-modal catastrophic forgetting does indeed occur in UMGMs when they are continually fine-tuned on a sequence of multimodal understanding tasks. We attribute this

forgetting to the UMGMs utilizing a unified architecture (e.g., transformer) with shared parameters for both text and visual modalities [61]. When tuning on a single modality (e.g., text), the shared parameters are updated in a way that interferes with the model's performance on the other modality (e.g., image), a phenomenon consistent with observations reported in recent studies [62]. To formalize this interference, we introduce the notion of *modality gradient conflict*.

**Definition 1** (Modality Gradient Conflict). Let $g_v = \nabla_\theta \mathcal{L}_v$ denote the gradient of the image-generation loss and $g_t = \nabla_\theta \mathcal{L}_t$ denote the gradient of the text-generation loss, both with respect to shared model parameters $\theta$. $g_v$ and $g_t$ are said to be conflicting with each other if $\langle g_v, g_t \rangle < 0$, where $\langle \cdot, \cdot \rangle$ denotes the standard Euclidean inner product.

**Proposition 1.** *When fine-tuning on the text-generation tasks, a stochastic gradient descent (SGD) update with sufficiently small step size $\eta$ modifies the model parameters as $\theta \leftarrow \theta - \eta g_t$. The resulting change in the visual loss is:*

$$\Delta \mathcal{L}_v = \mathcal{L}_v(\theta - \eta g_t) - \mathcal{L}_v(\theta) = -\eta \langle g_t, g_v \rangle + \frac{\eta^2}{2} g_t^\top H_v g_t, \tag{1}$$

*where $H_v = \nabla_\theta^2 \mathcal{L}_v$ is the Hessian of the visual loss. Hence, when $\langle g_t, g_v \rangle < 0$, a step optimizing text generation will* increase *the visual loss, leading to inter-modal forgetting.*

Motivated by this, we propose **MoDE**, a modality-decoupled adapter framework that provably eliminates the first-order modality conflict and thereby mitigates the inter-modal catastrophic forgetting. A detailed proof of Proposition 1 is available in Appendix A.

# 4 Methodology

## 4.1 Problem formulation

**Continual instruction tuning** Assume a UMGM has been pre-trained with abundant vision-language data, captured by trainable parameters $\theta$. We further train to adapt this UMGM to new $S$ tasks in a sequential manner. Each task is denoted by a task descriptor $\tau \in \{1, 2, ..., S\}$, and owns an independent dataset $D_\tau = \left\{ (X_{\tau,j}^{img}, X_{\tau,j}^{ins}, X_{\tau,j}^{ans}) \right\}_{j=1}^{N_\tau}$ with $N_\tau$ data triplets, where $X^{img}$, $X^{ins}$ and $X^{ans}$ indicate the input image tokens, instruction text tokens, and answer text tokens, respectively.

We formalize the autoregressive cross-entropy objective for task $\tau$ as:

$$\max_\theta \mathbb{E}_{(x^{img}, x^{ins}, x^{ans}) \sim \mathcal{D}_\tau} \sum_{i=1}^{L} \log p_\theta(X_i^{ans} \mid X^{img}, X^{ins}, X_{<i}^{ans}) \tag{2}$$

where the expectation $\mathbb{E}$ is taken over the data distribution $\mathcal{D}_\tau$, $L$ is the length of the answer sequence, $X_{<i}^{ans}$ indicate all answer tokens before the index $i$, $X_i^{ans}$ indicates the $i$-th answer token.

**Image generation** Let $X^{ins}$ be the text prompt tokens, and let $X^{img}$ be the image tokens generated autoregressively. Following a standard autoregressive factorization, the process can be formalized as:

$$p(X^{img} \mid X^{ins}) = \prod_{i=1}^{L} p_\theta(X_i^{img} \mid X^{ins}, X_{<i}^{img}) \tag{3}$$

where $L$ is the length of the image token sequence, $X_{<i}^{img}$ indicate all image tokens before the index $i$, $X_i^{img}$ indicate the $i$-th image token. Once all tokens are generated, they are detokenized via a decoder (such as a VQ-VAE) to produce the resultant image.

## 4.2 Modality-Decoupled Experts (MoDE)

MoE adapters are lightweight modules that route inputs through a sparse subset of expert networks, enabling task-specific adaptation with minimal interference to shared parameters [63]. Although MoE adapters can effectively mitigate *intra-modal* catastrophic forgetting, naively sharing their parameters across modalities can lead to modality gradient conflict (see Section 3). To address this issue, we propose **Modality-Decoupled Experts (MoDE)**, a modality-decoupled adapter architecture that

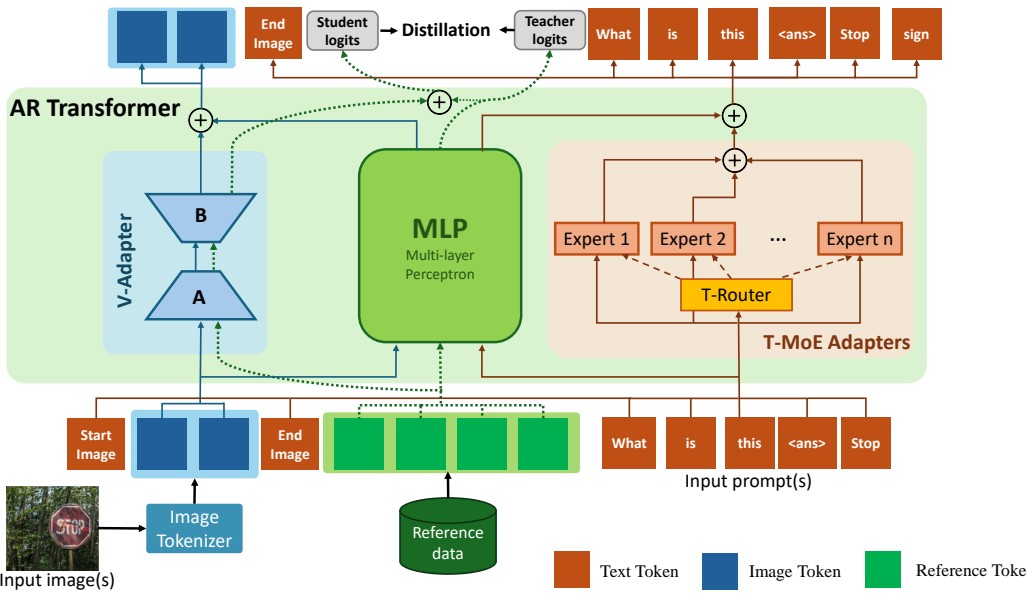

Figure 3: An autoregressive UMGM with our proposed MoDE integrated into its linear layers (MLPs). **V-Adapter** (Visual LoRA, in the light blue box): LoRA specialized for both the generation and understanding of image tokens. **T-MoE Adapters** (Text Mixture-of-Experts LoRA, in the light brown box): MoE-LoRA designed for text tokens, supporting continual learning of multimodal understanding tasks. **T-router** computes the routing weights $g_j(x)$ that determine how much each expert LoRA contributes for a given text token. The circled "+" symbol denotes addition. During continual instruction tuning, the T-MoE primarily updates for text answers, while the V-Adapter handles image tokens. To preserve the model's image generation capability and mitigate inter-modal forgetting, we apply a knowledge distillation loss from the original (teacher) UMGM to the new (student) model's V-Adapter.

isolates text and visual updates into separate trainable subspaces. As shown in Figure 3, MoDE mitigates modality gradient conflict while preserving the flexibility and scalability of MoE adapters.

Getting now into details, **LoRA** introduces a trainable low-rank update to a frozen weight matrix $W \in \mathbb{R}^{d_{\text{out}} \times d_{\text{in}}}$, where $d_{\text{in}}$ and $d_{\text{out}}$ denote the input and output feature dimensions, respectively. The update is parameterized by two learnable low-rank matrices $A \in \mathbb{R}^{r \times d_{\text{in}}}$ and $B \in \mathbb{R}^{d_{\text{out}} \times r}$, such that:

$$\Delta W = \frac{\alpha}{r} BA \tag{4}$$

where $A$ and $B$ are learnable matrices with rank $r \ll \min(d_{\text{in}}, d_{\text{out}})$ and scaling factor $\alpha \in \mathbb{R}$. Given an input token representation $h \in \mathbb{R}^{d_{\text{in}}}$, the modified linear transformation $f : \mathbb{R}^{d_{\text{in}}} \to \mathbb{R}^{d_{\text{out}}}$ becomes:

$$f(h) = hW^\top + \frac{\alpha}{r} hA^\top B^\top \tag{5}$$

**MoE-LoRA** generalizes LoRA by introducing a mixture of expert LoRA modules to handle diverse tasks. A router assigns each input token $x \in \mathbb{R}^{d_{\text{in}}}$ a soft distribution over experts:

$$g_j(x) = \text{softmax}(xW_g)_j \tag{6}$$

where $W_g \in \mathbb{R}^{d_{\text{in}} \times n}$ is a trainable gating matrix with $n$ denotes the number of experts, and $g_j(x)$ denotes the routing probability of expert $j$. The resulting update $\Delta W(x)$ is a weighted combination of expert LoRA updates:

$$\Delta W(x) = \frac{\alpha}{r} \sum_{j=1}^{n} g_j(x) B_j A_j \tag{7}$$

where each expert has $A_j \in \mathbb{R}^{r \times d_{\text{in}}}$ and $B_j \in \mathbb{R}^{d_{\text{out}} \times r}$, and $\alpha \in \mathbb{R}$ is a hyper-parameter. Given an input token representation $h \in \mathbb{R}^{d_{\text{in}}}$, the final linear transformation becomes:

$$f(h) = hW^\top + \frac{\alpha}{r} \sum_{j=1}^{n} g_j(x) hA_j^\top B_j^\top \tag{8}$$

**MoDE** integrates the above two types of adaptations into UMGMs: *V-Adapter* and *T-MoE*, as shown in Figure 3. The **V-Adapter** is a LoRA module specialized for visual tokens which adapts the model to both visual understanding and image generation tasks. In contrast, the **T-MoE** is a MoE-LoRA applied to text tokens, designed to enhance continual instruction tuning on multimodal understanding tasks without interfering with visual knowledge. Together, these two components allow MoDE to balance modality-specific adaptation (via the V-Adapter) and task-level flexibility (via the T-MoE) within a unified multimodal backbone.

To formalize the forward pass of multimodal tokens, we consider a frozen linear layer $W \in \mathbb{R}^{d_{\text{out}} \times d_{\text{in}}}$ in UMGM. Let $H = [h_1^t, \ldots, h_m^t, h_{m+1}^i, \ldots, h_{m+L}^i, h_{m+L+1}^t, \ldots, h_N^t] \in \mathbb{R}^{N \times d_{\text{in}}}$ denote the hidden states of a sequence of interleaved images and text with length $N$, where the subscript indicates position and the superscript indicates the token modality ($t$ for text and $i$ for image). We untie the sequence into the text sequence $H^t = [h_{m+L+1}^t, \ldots, h_N^t]$ and the image sequence $H^i = [h_{m+1}^i, \ldots, h_{m+L}^i]$, where $m$ is the start index of image tokens and $L$ is their length. We feed $H^i$ into the V-adapter using Eq. 5 to obtain $\hat{H}^i = f(H^i)$, and feed $H^t$ into T-MoE using Eq. 8 to obtain $\hat{H}^t = f(H^t)$. Finally, we reassemble $\hat{H}^i$ and $\hat{H}^t$ into their original positions to form the output sequence $\hat{H}$.

MoDE therefore cleanly decouples different modalities: text tokens benefit from experts diversity for continual instruction tuning, while image tokens rely on a visual adapter. Because T-MoE and V-adapter are disjoint, their update directions cannot interfere arbitrarily.

**Proposition 2.** *In MoDE, let $\phi$ denote the parameters of the T-MoE and $\psi$ denote the parameters of the V-adapter. A single gradient step $\theta \mapsto \theta - \eta \nabla_\phi \mathcal{L}_t$ updates only $\phi$, and the resulting change in the visual loss satisfies:*

$$\Delta \mathcal{L}_v = \frac{\eta^2}{2} \lambda_{\max}(\nabla_{\phi\phi}^2 \mathcal{L}_v) \|\nabla_\phi \mathcal{L}_t\|^2, \tag{9}$$

Thus, MoDE provably bounds the inter-modal interference to $\mathcal{O}(\eta^2)$, compared to $\mathcal{O}(\eta)$ under a modality-decoupled baseline as in Proposition 1. This theoretical property directly explains the minimal gradient conflict and superior retention of visual generation capabilities observed in our experiments. See Appendix A for a proof of Proposition 2 and Appendix B for its empirical verification.

### 4.3 Knowledge Distillation

During instruction tuning, the V-Adapter learns to *understand* images, which can degrade its original *generation* ability. We anchor the V-Adapter to the frozen pre-trained backbone (teacher) via logit-level knowledge distillation (KD), as shown in Figure 3.

A small subset of images sampled from the LAION-5B dataset [64] serves as the reference data. For each reference token, both teacher ($T$) and student ($S$) predict the next image token. Let $z_i^T, z_i^S$ denote their logits for the $i$-th image token. Using a softening factor $\beta$, the KD loss ($\mathcal{L}_{\text{KD}}$) is computed as the Kullback–Leibler divergence [65] between their softened output probabilities:

$$\mathcal{L}_{\text{KD}} = \beta^2 \sum_{i=1}^{L} D_{\text{KL}}\big(\text{Softmax}(z_i^T/\beta) \,\|\, \text{Softmax}(z_i^S/\beta)\big) \tag{10}$$

where $L$ is the number of generated image tokens and $\|$ denotes the distance between two distributions.

**Final objective.** The T-MoE is trained with the instruction-tuning cross-entropy loss ($\mathcal{L}_{\text{CE}}$) as in (2):

$$\mathcal{L}_{\text{T-MoE}} = \mathcal{L}_{\text{CE}} = -\frac{1}{L} \sum_{i=1}^{L} \log p_\theta(X_i^{ans} \mid X^{img}, X^{ins}, X_{<i}^{ans}) \tag{11}$$

The V-Adapter uses a weighted combination of the cross-entropy and the distillation losses:

$$\mathcal{L}_{\text{V-Adapter}} = \mathcal{L}_{\text{CE}} + \lambda \mathcal{L}_{\text{KD}}, \qquad \lambda > 0 \tag{12}$$

This mixed objective balances the updates for visual understanding and image generation in the V-adapter, thereby mitigating inter-modal catastrophic forgetting. In our experiments, we set $\lambda = 0.3$ based on its optimal performance (see Appendix D for a sweep over different $\lambda$ values).

Table 1: Quantitative comparison of inter-modal (image generation) and intra-modal (multimodal understanding) catastrophic forgetting. For image generation, Zero-shot reports the pre-trained reference we aim to preserve. DualPrompt [16] yields low accuracy and low forgetting. This is because adding a small set of prompt embeddings slightly biases the feature representations, which is insufficient to learn the new and hard tasks. $\Delta$ captures the overall performance drop. **Bold** numbers are best; underlined numbers are second best in each column. Results are averaged over three different runs.

| Method | Image Generation | | | Multimodal Understanding | | |
|---|---|---|---|---|---|---|
| | Text alignment ($\uparrow$) | Image alignment ($\uparrow$) | FID ($\downarrow$) | Accuracy ($\uparrow$) | Forgetting ($\downarrow$) | $\Delta$ ($\downarrow$) |
| Zero-shot | 0.2592 | 0.5205 | 52.13 | 22.48 | - | 34.84 |
| Seq LoRA | 0.2162 | 0.5150 | 56.12 | 28.43 | 35.33 | 28.57 |
| Model tailor [17] | 0.2384 | 0.5093 | 55.47 | 32.62 | 27.66 | 24.70 |
| DualPrompt [16] | **0.2648** | 0.5083 | 56.08 | 31.92 | **6.82** | 25.40 |
| MoELoRA [44] | 0.2248 | 0.5095 | 65.16 | 33.01 | 30.77 | 24.31 |
| CL-MoE [18] | 0.2081 | 0.5150 | 65.87 | 32.86 | 30.95 | 24.46 |
| **MoDE (Ours)** | 0.2458 | **0.5170** | **53.74** | **33.47** | 25.99 | **22.78** |

## 5 Experiments

### 5.1 Experimental setup

**Evaluation Benchmarks** For continual instruction tuning, we evaluate on five datasets: Four visual question answering datasets (ScienceQA [58], TextVQA [66], GQA [59], and VizWiz [67]) and one image classification dataset (ImageNet [60]). For image generation, we use the CustomConcept101 [68] dataset, which contains 101 concepts grouped into 16 broader categories.

**Evaluation Metrics** For continual instruction tuning, we follow the protocols from [69, 70] and report two metrics: (1) **Average Accuracy (ACC)**, defined as the mean accuracy across all tasks after the final round of training; and (2) **Average Forgetting (Fgt)**, which quantifies performance degradation on previous tasks and is computed as the mean reduction from the best observed accuracy to the final accuracy, for each task. We use exact matches to determine answers correctness.

We evaluate image generation quality with three metrics: (1) **Image alignment**. Cosine similarity between CLIP embeddings of the generated image and a reference image that depicts the target concept; higher is better ($\uparrow$). (2) **Text alignment**. Cosine similarity between the generated image embedding and the CLIP embedding of its text prompt; higher indicates better prompt fidelity ($\uparrow$). (3) **Fréchet Inception Distance (FID)** [71]. Fréchet distance between features of generated and real images; lower values mean the synthetic images are closer to the real distribution ($\downarrow$).

**Baselines** We compare our proposed MoDE with the following continual learning approaches: Seq LoRA, which sequentially trains the model on new tasks using LoRA adapters [39]; Model Tailor [17]; DualPrompt [16]; MoE-LoRA [63]; CL-MoE [18]; and Joint Training (upper bound), which trains the model on all datasets simultaneously and serves as the upper bound reference. The implementation details of baseline methods are provided in Appendix I.

We use two autoregressive backbone UMGMs: Chameleon [3] and Janus-Pro [4]. For Chameleon, we adopt the implementation in [72] since the original model and checkpoints are not publicly available. Results on Janus-Pro are provided in Appendix G. See Appendix D for implementation details.

### 5.2 Main results

**Quantitative results.** As shown in Table 1, our proposed MoDE built on Chameleon [3] effectively mitigates both inter- and intra-modal catastrophic forgetting, outperforming other CL baselines. In the "Image Generation" columns, MoDE maintains image quality comparable to the pre-trained model (Zero-shot), indicating strong resistance to inter-modal forgetting. For instance, MoDE achieves an FID score of 53.74, closely matching the 52.13 of the pre-trained model (zero-shot). While DualPrompt [16] also preserves image generation quality, its performance in continual instruction tuning (as shown in the "Multimodal Understanding" columns) is poor: the average accuracy is only 31.92%. Notably, although DualPrompt shows low average forgetting of 6.82%, this is misleading: its best accuracy on ImageNet is only 24.55%, whereas other methods exceed 70%. This low forgetting

Table 2: Continual instruction-tuning results on a sequence of five datasets. Accuracy (**ACC**) is exact-match (higher is better); **Fgt** is average forgetting (lower is better). For each method, the *upper row* shows the best accuracy during continual instruction tuning, and the *lower row* shows the final accuracy after completing all tasks. Upper bound reports performance obtained by individually fine-tuning a model on each task. **Bold** numbers denote the best result, and underlined numbers denote the second best. Results are averaged over three different runs.

| | Datasets | | | | | Metrics | |
|---|---|---|---|---|---|---|---|
| **Method** | ScienceQA | TextVQA | ImageNet | GQA | VizWiz | ACC (↑) | Fgt (↓) |
| Zero-shot | 51.52 | 23.49 | 16.53 | 14.22 | 6.64 | 22.48 | - |
| Seq LoRA | 72.43±0.82 | 39.34±1.24 | 89.41±1.48 | 37.93±1.61 | 44.38±1.05 | 28.43 | 35.33 |
| | 33.96±0.93 | 16.84±1.42 | 13.68±0.93 | 33.20±1.77 | 44.38±0.06 | | |
| Model tailor [17] | 74.90±0.72 | 40.04±1.18 | 78.22±0.63 | 37.35±1.02 | 43.25±0.94 | 32.62 | 27.66 |
| | 55.40±1.21 | 17.78±4.83 | 14.97±1.09 | 31.71±0.76 | 43.25±0.68 | | |
| DualPrompt [16] | 60.01±0.89 | 31.48±1.02 | 24.55±0.57 | 29.91±1.27 | 40.91±0.75 | 31.92 | **6.82** |
| | 59.64±1.03 | 19.68±3.81 | 8.61±0.09 | 30.75±0.69 | 40.91±0.72 | | |
| MoELoRA [44] | 71.79±0.94 | 39.62±2.08 | 94.75±1.53 | 37.66±1.23 | 44.33±0.65 | 33.01 | 30.77 |
| | 50.47±1.17 | 19.64±2.86 | 18.73±1.96 | 31.89±0.78 | 44.33±0.70 | | |
| CL-MoE [18] | 71.35±0.88 | 38.82±1.19 | 90.08±2.59 | 37.37±2.14 | 44.13±0.73 | 32.86 | 30.95 |
| | 55.34±1.09 | 17.64±2.77 | 15.62±1.02 | 31.57±0.81 | 44.13±0.67 | | |
| **MoDE (Ours)** | 70.45±0.74 | 38.90±1.47 | 80.67±0.45 | 37.03±0.47 | 44.35±0.62 | **33.47** | 25.99 |
| | 56.25±1.87 | 19.08±2.47 | 14.67±0.76 | 33.10±1.16 | 44.28±0.09 | | |
| Upper bound | 73.27 | 40.74 | 91.88 | 36.56 | 44.15 | 57.32 | - |

arises not from effective retention, but from a limited ability to learn in the first place: if a model never learns well, it has little to forget. Although MoE-LoRA [63] performs well in mitigating intra-modal forgetting, it suffers from severe inter-modal forgetting, as evidenced by the worse image generation quality, with an FID of 65.16 compared to 56.12 for the vanilla SeqLoRA. In contrast, our MoDE achieves the highest accuracy of 33.47% and the lowest FID of 53.74, demonstrating its effectiveness in mitigating both inter-modal and intra-modal forgetting simultaneously.

Table 2 presents more detailed results of continual instruction tuning. In this setting, our MoDE consistently outperforms all comparison methods and sustains effective learning across all tasks.

**Qualitative results.** As shown in Figure 4, our MoDE can generates higher quality images compared to other methods, effectively preserving the pre-trained model's image generation capability. For instance, in the third row of Figure 4, given the prompt "*Barn in the fall season with leaves all around*", MoDE successfully generates yellow leaves surrounding the barn, whereas Model Tailor [17] and CL-MoE [18] fail to capture this detail. See Appendix C for more qualitative results.

## 5.3 Ablation results

To validate the performance gains of MoDE, we present our ablation results in Table 3. "T-MoE LoRA" includes only the T-MoE adapters without the V-adapter. "MoDE w/o KD" removes knowledge distillation, training both the T-MoE adapters and the V-adapter solely with cross-entropy loss. Of note, MoDE is also robust to task order. See Appendix E for more ablation results.

Table 3: Ablation study shows that both the modality-specific adapters and knowledge distillation are essential to the effectiveness of MoDE.

| | Image Generation | | | Visual Understanding | |
|---|---|---|---|---|---|
| Model | Text alignment (↑) | Image alignment (↑) | FID (↓) | Accuracy (↑) | Forgetting (↓) |
| Chameleon [3] | 0.2592 | 0.5205 | 52.13 | 22.48 | - |
| + T-MoE LoRA | 0.2583 | 0.5317 | 51.28 | 33.03 | 28.65 |
| + MoDE w/o KD | 0.2364 | 0.5156 | 54.61 | 33.07 | 26.49 |
| + MoDE | **0.2458** | **0.5170** | **53.74** | **33.47** | **25.99** |

| Input Prompt | Chameleon [3] | Model Tailor [17] | CL-MoE [18] | MoDE (Ours) |
|---|---|---|---|---|
| *A dog wearing sunglasses on the porch.* | | | | |
| *A transparent cup filled with steaming hot cocoa.* | | | | |
| *Barn in the fall season with leaves all around.* | | | | |
| *Marigold flowers in the vase.* | | | | |

Figure 4: Qualitative results of image generation on the Chameleon [3] model. Our method generates more visually coherent and faithful images compared to other baselines (e.g., the realistic dog in the first row, steam in the second row). Additional examples are provided in Appendix C.

These results demonstrate the effectiveness of both the T-MoE and V-adapter. Using only the T-MoE preserves image generation capabilities by leaving visual components untouched, but results in sub-optimal visual understanding, as the model lacks updates necessary for better visual comprehension.

We also report the computational efficiency of MoDE, including training time, memory consumption, and parameter overhead, in Appendix H. These results show that MoDE achieves favorable trade-offs between accuracy and efficiency compared to existing methods.

## 6 Conclusion

We have proposed Modality-Decoupled Experts (MoDE) to address both intra- and inter-modal catastrophic forgetting in continual instruction tuning for Unified Multimodal Generative Models (UMGMs). We have identified inter-modal forgetting, explained its cause due to the modality gradient conflict, and designed MoDE to decouple modality-specific updates, mitigating conflict theoretically and empirically. Extensive experiments show that MoDE significantly outperforms SOTA baselines, establishing it as a strong solution for continual learning in unified multimodal generation.

## 7 Acknowledgments

This work is supported in part by the NSF grant CCF-2531882, and in part by an Amazon Research Award, Spring 2025.

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

# A Theoretical Analysis

We formally compare the one-step visual loss drift between a *modality-coupled* architecture (shared parameters for text and image) and our *modality-decoupled* MoDE design. Let assume the following parameters:

- $\theta$ the full set of trainable parameters,
- $\mathcal{L}_t(\theta)$ the text-generation loss,
- $\mathcal{L}_v(\theta)$ the image-generation loss,
- $\eta$ the learning rate,
- $g_t = \nabla_\theta \mathcal{L}_t$ and $g_v = \nabla_\theta \mathcal{L}_v$ the corresponding gradients of text and visual modalities, respectively,
- $H_v = \nabla_\theta^2 \mathcal{L}_v$ the Hessian of the visual loss.

After a gradient step $\theta \leftarrow \theta - \eta g_t$, the change in the visual loss is, by a second-order Taylor expansion:

$$\Delta \mathcal{L}_v = \mathcal{L}_v(\theta - \eta g_t) - \mathcal{L}_v(\theta) = -\eta \langle g_t, g_v \rangle + \frac{\eta^2}{2} g_t^\top H_v g_t + o(\eta^2) \tag{13}$$

where $o(\eta^2)$ denotes higher-order terms that vanish faster than $\eta^2$.

**Modality-coupled architecture (shared parameters).** In the modality-coupled case (e.g., shared MoE-LoRA adapters between modalities), both $g_t$ and $g_v$ are nonzero over the same parameter space. Thus, the first-order term $-\eta\langle g_t, g_v \rangle$ is generally nonzero. In the worst case (maximum interference), $g_t$ and $g_v$ are anti-aligned, so by Cauchy–Schwarz inequality we have:

$$\boxed{|\Delta \mathcal{L}_v^{\text{coupled}}| \leq \eta \, \|g_t\| \, \|g_v\| + \frac{\eta^2}{2} \lambda_{\max}(H_v) \|g_t\|^2 + o(\eta^2)} \tag{14}$$

Thus, the dominant error scales as $\mathcal{O}(\eta)$.

**Modality-decoupled architecture (MoDE).** In MoDE, the text and visual tokens update *disjoint* parameter blocks $\phi$ and $\psi$, respectively; thus:

$$\frac{\partial \mathcal{L}_v}{\partial \phi} = 0, \qquad \frac{\partial \mathcal{L}_t}{\partial \psi} = 0,$$

which implies

$$\langle g_t, g_v \rangle = 0 \quad \text{(orthogonal supports)}$$

Therefore, the first-order term in (13) vanishes, and the remaining visual loss drift is purely second-order:

$$\boxed{|\Delta \mathcal{L}_v^{\text{MoDE}}| \leq \frac{\eta^2}{2} \lambda_{\max}(H_v) \|g_t\|^2 + o(\eta^2)} \tag{15}$$

Thus, the dominant error scales as $\mathcal{O}(\eta^2)$.

**Conclusion.** Comparing(14) and(15), we conclude that

$$|\Delta \mathcal{L}_v^{\text{MoDE}}| \ll |\Delta \mathcal{L}_v^{\text{coupled}}|$$

since $\eta^2 \ll \eta$ for small $\eta$.

Thus, **MoDE provably limits the worst-case drift in image-generation loss to** $\mathcal{O}(\eta^2)$, compared to $\mathcal{O}(\eta)$ in a modality-coupled architecture.

# B  Modality gradient conflict

The theoretical support is provided in Appendix A. This section provides empirical evidence (Figure 5, 6) for our claim.

As shown in Figure 5, the modality-coupled MoE LoRA [44] displays significant gradient conflict between text and image modalities, empirically validating our analysis in Section 3.

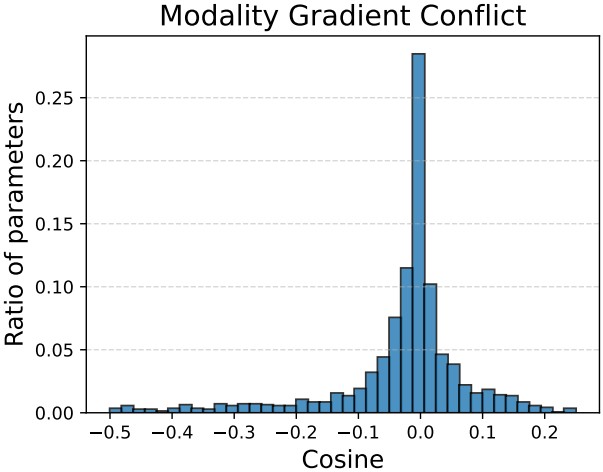

Figure 5: Cosine distance distribution between text and image modality gradients in modality-coupled MoE LoRA [44] on the ScienceQA [58] dataset. The y-axis shows the proportion of parameters corresponding to each cosine distance. Lower cosine distance values indicate greater gradient conflict, with 0 denoting orthogonal update directions.

In contrast to the significant conflict shown in Figure 5, Figure 6 clearly shows that MoDE has zero cosine disagreement between modalities. This confirms that its modality-specific update subspaces are completely isolated, both theoretically and empirically.

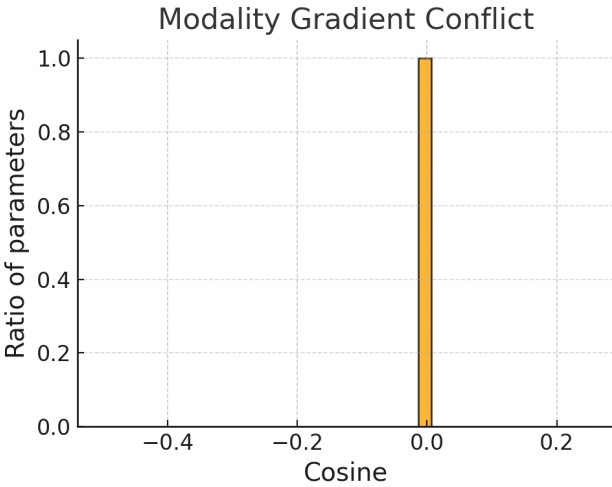

Figure 6: Cosine distance distribution between text and image modality gradients in MoDE. Lower cosince distance values indicate greater gradient conflict, with 0 denoting orthogonal directions. In MoDE, all parameters exhibit perfectly orthogonal gradients, confirming the absence of modality gradient conflict.

# C   Additional Qualitative Results

Table 4 presents the complete qualitative results of MoDE, built on Chameleon [3], in comparison to various baseline methods. Our approach generates images that better align with the prompts and exhibit higher visual fidelity. For example, given the prompt "*Barn in the fall season with leaves all around*", only the original Chameleon and our MoDE successfully generate images featuring yellow leaves around the barn.

Table 4: Full qualitative results of our MoDE compared to other baseline methods.

| Method | *A dog wearing sunglasses on the porch* | *A transparent cup filled with steaming hot cocoa* | *Barn in the fall season with leaves all around* | *Marigold flowers in the vase* |
|---|---|---|---|---|
| Chameleon [3] | | | | |
| Seq LoRA | | | | |
| Model Tailor [17] | | | | |
| DualPrompt [16] | | | | |
| MoELoRA [44] | | | | |
| CL-MoE [18] | | | | |
| **MoDE (Ours)** | | | | |

# D   Implementation Details

We use Chameleon [3] and Janus-Pro [30] as backbone models. For Chameleon, we adopt the implementation from [72] due to the unavailability of the original checkpoints and code. All LoRA modules are configured with a default rank of 8. Training is conducted for one epoch per task on 8 NVIDIA RTX 6000 GPUs, with bf16 mixed precision and a per-GPU batch size of 1 (total effective batch size of 8).

We use a learning rate of $1 \times 10^{-4}$, scheduled via cosine decay with a warm-up ratio of 0.1. The softening factor $\beta$ in knowledge distillation equation (10) is 2.0. The hyperparameter $\lambda$ is set to 0.3, selected based on a sweep over multiple values (see Table 5). This value strikes a balance between image generation and visual understanding: higher $\lambda$ favors generation quality, while lower $\lambda$ improves understanding performance. A higher $\lambda$ increases the weight of the knowledge distillation loss in the V-adapter (see Eq. 12), constraining the update closer to the pre-trained model to preserve image generation capabilities. However, this also constrains beneficial updates needed for improved visual understanding.

Table 5: Performance of MoDE under different values of the loss balancing coefficient $\lambda$. We select $\lambda = 0.3$ (bolded) as it yields the best balance between image generation fidelity and visual understanding accuracy.

| | Image Generation | | | Visual Understanding | |
|---|---|---|---|---|---|
| $\lambda$ | Text alignment ($\uparrow$) | Image alignment ($\uparrow$) | FID ($\downarrow$) | Accuracy ($\uparrow$) | Forgetting ($\downarrow$) |
| 0.0 | 0.2364 | 0.5156 | 54.61 | 33.07 | 26.49 |
| **0.3** | **0.2458** | **0.5170** | **53.74** | **33.47** | **25.99** |
| 0.5 | 0.2498 | 0.5264 | 53.12 | 31.53 | 28.38 |
| 1.0 | 0.2543 | 0.5190 | 51.72 | 32.10 | 27.62 |

# E   Additional Ablation Results

## E.1   Different Tasks Order

Following [63], we explore the impact of different tasks order by conducting an additional experiment using a different order of tasks, arranged alphabetically: GQA [59], ImageNet [60], ScienceQA [58], TextVQA [66], and VizWiz [67].

Table 6: Results of inter-modal (image generation) and intra-modal (multimodal understanding) catastrophic forgetting on different task orderings. For image generation, Zero-shot reports the pre-trained reference we aim to preserve. **Bold** shows the best results, underline shows the second best results. Chameleon [3] is used for the backbone.

| | **Image Generation** | | | **Multimodal Understanding** | |
|---|---|---|---|---|---|
| **Method** | Text alignment ($\uparrow$) | Image alignment ($\uparrow$) | FID ($\downarrow$) | Accuracy ($\uparrow$) | Forgetting ($\downarrow$) |
| Zero-shot | 0.2592 | 0.5205 | 52.13 | 22.48 | - |
| Seq LoRA | 0.2307 | 0.4973 | 66.75 | 27.72 | 33.10 |
| Model tailor [17] | 0.2448 | 0.5166 | 53.58 | 34.02 | 24.51 |
| DualPrompt [16] | 0.2491 | 0.5140 | 53.07 | 32.05 | **7.96** |
| MoELoRA [63] | 0.2492 | 0.5125 | 61.64 | 34.41 | 24.72 |
| CL-MoE [18] | 0.2463 | 0.5090 | 59.89 | 34.18 | 25.63 |
| **MoDE (Ours)** | **0.2525** | **0.5256** | **51.69** | **36.68** | 22.92 |

As shown in Table 6, our MoDE consistently outperforms the baseline methods, thus demonstrating its robustness to different task orderings.

## E.2   Different Number of Experts

This this section we provide an ablation study on the number of experts in the T-MoE module.

Table 7: Performance of MoDE under different number of experts in T-MoE.

| | Image Generation | | | Visual Understanding | |
|---|---|---|---|---|---|
| # Experts | Text alignment (↑) | Image alignment (↑) | FID (↓) | Accuracy (↑) | Forgetting (↓) |
| 2 | 0.2437 | 0.5144 | 53.61 | 32.24 | 26.81 |
| 4 | 0.2458 | 0.5170 | 53.74 | 33.47 | 25.99 |
| 6 | 0.2499 | 0.5146 | 52.69 | 34.90 | 24.12 |

As shown in Table 7, increasing the number of experts yields slightly improved text alignment and accuracy, though the gains are modest. This suggests that the performance of MoDE is relatively robust to the number of experts, and 4 experts offer a good trade-off between performance and computational cost.

### E.3 Ablation on Knowledge Distillation and Modality Decoupling

We provide the results of adding our cross-modal knowledge distillation (KD) loss to modality-coupled baselines (SeqLoRA [39], CL-MoE [18], MoELoRA [63]) in Table 8. Although applying KD to vanilla LoRA leads to modest improvements both in image generation quality and multimodal understanding performance, we observe that for MoELoRA and CL-MoE this mitigates visual-generation forgetting at the cost of hurting the multimodal understanding performance. This shows that without explicit modality decoupling, KD "locks in" the teacher's features in a way that hinders the model's ability to adapt to new multimodal understanding tasks. The results come from a single run.

Table 8: Effect of adding KD to modality-coupled baselines.

| | Image Generation | | | Visual Understanding | |
|---|---|---|---|---|---|
| | Text alignment (↑) | Image alignment (↑) | FID (↓) | Accuracy (↑) | Forgetting (↓) |
| Seq LoRA [39] | 0.2162 | 0.5150 | 56.12 | 28.43 | 35.33 |
| SeqLoRA w/ KD | **0.2478** | **0.5180** | **52.83** | **30.81** | **29.85** |
| MoELoRA [44] | 0.2248 | 0.5095 | 65.16 | **33.01** | **30.77** |
| MoELoRA w/ KD | **0.2559** | **0.5176** | **53.03** | 30.07 | 35.44 |
| CL-MoE [18] | 0.2081 | 0.5150 | 65.87 | **32.86** | **30.95** |
| CL-MoE w/ KD | **0.2439** | **0.5200** | **52.43** | 31.82 | 34.74 |

As shown in Table 9, decoupling textual and visual LoRA without knowledge distillation yields only modest gains in understanding accuracy and visual generation quality compared to vanilla LoRA. This indicates that decoupling alone is insufficient to maintain the rich cross-modal alignment necessary for high-quality generation. Furthermore, Table 3 includes an ablation on "MoDE w/o KD" (i.e., decoupled T-MoE and V-adapter without knowledge distillation), which further confirms that knowledge distillation plays a critical role in the performance improvements observed.

Table 9: Ablation on modality-decoupled architecture.

| | Image Generation | | | Visual Understanding | |
|---|---|---|---|---|---|
| | Text alignment (↑) | Image alignment (↑) | FID (↓) | Accuracy (↑) | Forgetting (↓) |
| Seq LoRA | 0.2162 | 0.5150 | 56.12 | 28.43 | 35.33 |
| Decoupled LoRA | 0.2252 | 0.5180 | 54.74 | 30.59 | 32.79 |

The ablation results show that MoDE is more than the sum of its parts: only the combination of modality-decoupling plus knowledge distillation leads to consistently superior performance.

## F    Multimodal Understanding Forgetting

In this section, we provide the results of MoDE mitigating catastrophic forgetting of the understanding capabilities induced by fine-tuning UMGMs on image generation tasks. To address this, we conduct new experiments by fine-tuning the linear layers of the Chameleon model [3] on image generation tasks. Specifically, we use the LAION-5B dataset [64] for fine-tuning and apply knowledge distillation using the multimodal understanding sequence from the GQA dataset [59] as reference data. We compare our MoDE method with LoRA fine-tuning across three multimodal understanding benchmarks. Accuracies are reported in Table 10. The difference between zero-shot and fine-tuned performance reflects the degree of forgetting in multimodal understanding.

Table 10: Forgetting caused by fine-tuning on image generation tasks.

| Method | Multimodal Understanding Accuracy (%) | | |
| --- | --- | --- | --- |
| | ScienceQA [58] | TextVQA [66] | ImageNet [60] |
| Zero-shot | 51.52 | 23.49 | 16.53 |
| LoRA [39] | 45.81 | 12.25 | 0.00 |
| **MoDE (Ours)** | **51.28** | **23.44** | **16.61** |

The results demonstrate that fine-tuning on image generation tasks can indeed cause catastrophic forgetting in multimodal understanding. This is evident because of the substantial drop in performance for the LoRA baseline compared to zero-shot. Our MoDE method remains effective in this scenario, significantly reducing the performance drop and mitigating forgetting relative to the LoRA baseline. This demonstrates that MoDE generalizes beyond the specific forgetting direction explored in the main paper, where understanding tasks overwrite prior image generation capabilities, and is effective regardless of the fine-tuning direction.

## G    Additional Continual Learning Results

### G.1    Additional Results with Janus-Pro

We provide additional quantitative comparisons using Janus-Pro-1B [30] as the backbone. We further include the Grounding task with referring expression grounding datasets (RefCOCO [73], RefCOCO+ [74], and RefCOCOg [74]). All methods are trained with early stopping. As shown in Table 11, our MoDE outperforms the baseline methods in continual instruction tuning for multimodal understanding.

Table 11: Results of continual instruction tuning with Janus-Pro-1B [30] as the backbone. Accuracy (**ACC**) is exact-match (higher is better); **Fgt** is average forgetting (lower is better). For each method, the *upper row* shows the best accuracy during continual instruction tuning, and the *lower row* shows the final accuracy after completing all tasks. Upper bound reports performance obtained by individually fine-tuning a model on each task. **Bold** shows the best results.

| Method | Datasets | | | | | | Metrics | |
| --- | --- | --- | --- | --- | --- | --- | --- | --- |
| | ScienceQA | TextVQA | ImageNet | GQA | VizWiz | Grounding | ACC (↑) | Fgt (↓) |
| Zero-shot | 82.69 | 40.22 | 13.11 | 38.46 | 26.74 | 0.00 | 35.54 | - |
| Seq LoRA | 61.27 | 51.34 | 96.89 | 58.21 | 56.33 | 41.65 | 37.65 | 27.37 |
| | 44.87 | 32.01 | 12.42 | 47.25 | 47.71 | 41.65 | | |
| Model Tailor [17] | 61.58 | 47.36 | 92.79 | 51.48 | 44.87 | 36.72 | 29.13 | 32.01 |
| | 38.62 | 35.76 | 21.18 | 32.98 | 9.49 | 36.72 | | |
| CL-MoE [18] | 74.17 | 50.24 | 93.21 | 56.86 | 55.52 | 38.92 | 43.18 | 21.96 |
| | 44.22 | 42.38 | 34.63 | 55.04 | 43.92 | 38.92 | | |
| **MoDE (Ours)** | 81.01 | 46.34 | 95.16 | 62.01 | 61.02 | 44.99 | **53.02** | **14.48** |
| | 65.45 | 39.10 | 52.32 | 58.20 | 58.07 | 44.99 | | |
| Upper bound | 86.28 | 51.34 | 98.72 | 62.55 | 67.23 | 46.13 | 68.71 | - |

## G.2 Additional Results with Chameleon

Following the CoIN benchmark [63], we present results on additional datasets using Chameleon [3] as the backbone. We further include the Grounding task with referring expression grounding datasets (RefCOCO [73], RefCOCO+ [74], and RefCOCOg [74]). All methods are trained with early stopping. As shown in Table 12, our MoDE outperforms the baseline methods in continual instruction tuning for multimodal understanding.

Table 12: Results on a sequence of six datasets. Accuracy (**ACC**) is exact-match (higher is better); **Fgt** is average forgetting (lower is better). For each method, the *upper row* shows the best accuracy during continual instruction tuning, and the *lower row* shows the final accuracy after completing all tasks. Upper bound reports performance obtained by individually fine-tuning a model on each task. **Bold** shows the best results.

| | Datasets | | | | | | Metrics | |
| Method | ScienceQA | TextVQA | ImageNet | GQA | VizWiz | Grounding | ACC (↑) | Fgt (↓) |
|---|---|---|---|---|---|---|---|---|
| Zero-shot | 51.52 | 23.49 | 16.53 | 14.22 | 6.64 | 0.00 | 18.73 | - |
| Seq LoRA | 72.43 | 39.34 | 89.41 | 37.93 | 44.38 | 18.62 | 29.82 | 24.64 |
| | 38.67 | 33.18 | 15.09 | 34.99 | 39.34 | 18.62 | | |
| Model tailor [17] | 74.90 | 40.04 | 78.22 | 37.35 | 43.25 | 15.66 | 29.09 | 22.97 |
| | 56.06 | 25.82 | 7.10 | 32.62 | 37.32 | 15.66 | | |
| DualPrompt [16] | 60.01 | 31.48 | 24.55 | 29.91 | 40.91 | 13.82 | 25.47 | **9.57** |
| | 52.45 | 24.12 | 3.82 | 31.44 | 27.16 | 13.82 | | |
| MoELoRA [44] | 71.79 | 39.62 | 94.75 | 37.66 | 44.33 | 37.13 | 34.14 | 24.08 |
| | 65.89 | 22.36 | 6.71 | 30.22 | 42.55 | 37.13 | | |
| CL-MoE [18] | 71.35 | 38.82 | 90.08 | 37.37 | 43.73 | 36.22 | 36.96 | 19.12 |
| | 67.32 | 32.82 | 9.12 | 33.29 | 43.20 | 36.22 | | |
| **MoDE (Ours)** | 71.49 | 39.00 | 85.88 | 36.73 | 44.45 | 36.64 | **37.05** | 18.25 |
| | 67.87 | 33.36 | 9.54 | 33.60 | 41.91 | 36.64 | | |
| Upper bound | 73.27 | 40.74 | 91.88 | 36.56 | 44.15 | 44.90 | 55.25 | - |

# H  Computational Analysis

In this section, we report comparisons of training compute, memory footprint, and trainable parameter size.

Table 13: Comparison of trainable parameters and training costs. Results come from a single run.

| Method | TFLOPs | Peak GPU Memory (MB) | Training Time (ms) | Trainable Params (%) |
|---|---|---|---|---|
| MoELoRA [63] | 4.29 | 75265 | 419.2 | 0.0171 |
| CL-MoE [18] | 4.29 | 86300 | 421.7 | 0.0171 |
| **MoDE (Ours)** | 4.19 | 84980 | 440.3 | 0.0211 |

As shown in Table 13, MoDE introduces only a slight increase in training time (440.3 ms vs. 419.2 ms) and peak GPU memory (84.98 GB vs. 75.27 GB) compared to MoELoRA [63], while maintaining comparable TFLOPs. The trainable parameter ratio of MoDE is only 0.0211%, modestly higher than MoELoRA [63] and CL-MoE [18] (0.0171%), indicating that our performance gains are not simply due to increased processing capacity. These results suggest that MoDE's improvements arise from its design (modality-decoupled experts and distillation), rather than significantly higher resource usage.

# I  Baseline Settings

In this section, we provide the experimental settings for the baseline methods used in our experiments.

## I.1 Overview of Baselines

- **Model Tailor** [17] mitigates catastrophic forgetting in MLLMs by (i) identifying a sparse "model patch" via a fused salience-and-sensitivity score, and (ii) applying Hessian-based weight compensation so the patch can be safely merged back into the frozen backbone.

- **DualPrompt** [16] is a replay-free continual learning method grafting two tiny prompt sets onto a frozen transformer backbone: a shared G-Prompt that encodes task-invariant knowledge and per-task E-Prompts keyed to input features for task-specific skills.

- **MoELoRA** [44] extends LoRA by substituting each single low-rank adapter with a *mixture of N experts*. The LoRA update is rewritten as:

$$h = Wx + \tfrac{\alpha}{r} \sum_{i=1}^{N} G_i B_i A_i x,$$

where every expert has rank $r/N$ and is gated by $G_i$, so the *total* trainable parameters remain identical to vanilla LoRA while the separate experts capture more diverse, task-specific representations.

- **CL-MoE** [18] tackles continual VQA on top of an MLLM by (i) introducing a *Dual-Router MoE*: a task-level router picks global experts while an instance-level router refines the choice per example, and (ii) applying a *dynamic momentum update* that blends new LoRA-based expert weights with the frozen past experts, so the model absorbs new skills yet preserves prior knowledge, sharply reducing catastrophic forgetting.

- **Seq LoRA** is a vanilla baseline that applies LoRA adapters in plain sequential fine-tuning across tasks.

## I.2 Training Settings

To ensure a fair comparison, we follow the hyperparameters and implementation details from each baseline's original codebase:

- **Model Tailor** [17]: sparsity level set to 10% (percentage of parameters replaced).

- **DualPrompt** [16]: prompt lengths $L_g = 5$ and $L_e = 20$; prompts are injected at layers $\texttt{start}_g = 1$, $\texttt{end}_g = 2$, $\texttt{start}_e = 3$, and $\texttt{end}_e = 5$.

- **MoE-LoRA** [44]: number of experts set to 4, with each expert having LoRA rank 8.

- **CL-MoE** [18]: number of experts set to 4, with each expert having LoRA rank 8. Top-$K$ experts selected with $K = 2$. Hyperparameters $\gamma = 0.7$, $\beta = 0.5$.

# J  Ethics Statement

This work does not involve human subjects, personally identifiable information, or proprietary data. All datasets used (such as ScienceQA, TextVQA, GQA, VizWiz, and ImageNet) are publicly available and commonly used in the research community. We follow the licensing and usage policies associated with each dataset. Our continual learning method, MoDE, improves the robustness of unified multimodal generative models without modifying or exploiting sensitive data.

As with all powerful generative models capable of producing synthetic images from text, our method carries a potential risk of misuse, such as generating disinformation or harmful visual content. Although MoDE focuses on mitigating forgetting in multimodal learning, enhanced generation capabilities may inadvertently facilitate deceptive or malicious use. We recommend that practitioners deploying such models consider safeguards such as content filtering, provenance tracking, and visible or invisible watermarking of generated content to support responsible use and attribution.

