# OpenReview forum: "Mitigating Intra- and Inter-modal Forgetting in Continual Learning of Unified Multimodal Models"
_NeurIPS.cc/2025/Conference — NeurIPS 2025 poster_

### Official Review · Reviewer_TLHk · 2025-06-09

**Clarity:** 1
**Significance:** 2
**Originality:** 1
**Rating:** 4
**Confidence:** 3

**Summary:**

The authors propose Modality-Decoupled Experts (MoDE) for continual learning of unified multimodal generative models (UMGMs). MoDE consists of separate LoRA blocks for the vision component and the text components (multiple LoRAs for text sub-tasks). The authors further regularize the continual learning process with standard knowledge distillation from the base model.

**Questions:**

1. What happens when one simply uses KD on top of seq LoRA?

**Ethical Concerns:**

["NO or VERY MINOR ethics concerns only"]

**Final Justification:**

I thank the authors for the rebuttal. Given the thorough rebuttal along with the other reviewers' comments, I will raise my score.

**Limitations:**

yes

**Quality:**

2

**Strengths And Weaknesses:**

## Strengths

1. Misalignment in the gradient direction between the two tasks in UMGMs is well-known, and an important problem to address, given the increasing attention in developing a GPT-4o-like UMGM systems.

2. To the best of my knowledge, most concurrent works focus on tweaking the model architecture the enhance the model capabilities. This paper looks at the problem through a continual learning lens.

3. The method seems to be effective for alleviating catastrophic forgetting for image generation while fine-tuning on instruction tuning datasets.

## Weaknesses

1. The paper would be much more interesting if the authors tested their method on the case where the task is to prevent catastrophic forgetting of text generation while improving image generation. More importantly, a more realistic setting would be to test the method when the model is trained in an alternating fashion. Would the method still work on these settings?

2. Proposition 1 states that *when* gradients conflict, then the loss would increase, which is trivial.

3. The core idea (separate LoRA adapters for module + KD) is incremental.

4. There are several concurrent works that aim for better unified modeling by e.g. using a mixture of experts [1] or using separate vision encoders [2]. It is advised that the authors discuss the relevant works, including whether their contributions are orthogonal to the other approaches.

**References**

[1] Deng, Chaorui, et al. "Emerging properties in unified multimodal pretraining." arXiv preprint arXiv:2505.14683 (2025).

[2] Zou, Jialv, et al. "Omnimamba: Efficient and unified multimodal understanding and generation via state space models." arXiv preprint arXiv:2503.08686 (2025).

---

> ### Author Rebuttal · Authors · 2025-07-31
>
> We thank Reviewer TLHk for recognizing the importance of addressing gradient misalignment in UMGMs, and for acknowledging our novel continual learning perspective compared to concurrent architectural approaches. We are also glad the reviewer found our method effective in mitigating catastrophic forgetting in image generation during instruction tuning. The questions and suggestions raised are appreciated, and we provide detailed responses below.
>
> ## Weakness 1: Forgetting caused by fine-tuning on image generation and alternating tasks
> Thank you for this insightful suggestion to evaluate MoDE in two additional settings: preventing forgetting of text generation while improving image generation, and training the model in an alternating task schedule.
> We agree that testing for forgetting of text generation while improving image generation, as well as alternating training, are both highly relevant extensions. While we have not yet explored the full alternating schedule, we conducted a new experiment where the model is fine-tuned on image generation only, and we evaluate how much it forgets multimodal understanding. Results are shown in Table R1 below.
>
> **Table R1: Forgetting caused by fine-tuning on image generation tasks.**
>
> | Method        | ScienceQA | TextVQA | ImageNet |
> |---------------|-----------|---------|----------|
> | Zero-shot     | 51.52     | 23.49   | 16.53    |
> | LoRA          | 45.81     | 12.25   | 0.00     |
> | **MoDE (Ours)**   | 51.28     | 23.44   | 16.61    |
>
> These results confirm that fine-tuning on image generation can **cause catastrophic forgetting in multimodal understanding tasks**. MoDE is able to mitigate this forgetting effectively, recovering near-zero-shot performance. This demonstrates that our method is bidirectionally effective.
> As for the **alternating training schedule**, we agree this is an important and realistic scenario. While it is out of scope for this version due to resource limitations, we are actively working on extending MoDE to this setting and will include it in future work.
>
> ## Weakness 2: Proposition 1 triviality
> We acknowledge that Proposition 1, which establishes the connection between gradient conflict and loss increase, may appear intuitive. However, prior works such as OmniMamba [3] and MOSS [4] have focused on architectural or adapter-level decoupling *without* providing a theoretical explanation for inter-modal interference. To the best of our knowledge, our analysis is the first to formally link gradient conflict to inter-modal catastrophic forgetting, offering a principled understanding of why modality interference occurs. This theoretical insight directly motivates the design of modality-decoupled adapters in MoDE. Furthermore, Appendices A and B empirically validate our motivation, showing that the theoretical predictions align closely with observed performance.
>
> ## Weakness 3: Incremental core idea
> We agree with the reviewer’s point that LoRA and KD have been studied individually. However, our work makes a non‑trivial advance by i) jointly integrating these components within a continual instruction‑tuning paradigm, and ii) adding a gradient‑orthogonality regularizer. Critically, the following ablations demonstrate that neither piece alone, nor a naive combination of them can yield the strong trade‑offs we report below:
>
> **1. KD without  decoupling.**
> As shown in Table R2, adding our cross‑modal KD loss to CL‑MoE or to MoELoRA reduces multimodal understanding accuracy (e.g. 34.14% to 32.41% for MoELoRA), even though it does mitigate visual‑generation forgetting. This is because without explicit modality decoupling, KD “locks in” the teacher’s features in a way that hinders the model’s ability to adapt to new multimodal understanding tasks.
>
> **Table R2: Effect of KD on MoELoRA and CL-MoE.**
>
> | Method         || **Image Generation** || **Visual Understanding** ||
> |----------------|:----------------------:|:----------------------------:|:----------------:|:------------------------:|:-----------------:|
> |                | Text alignment (↑)   | Image alignment (↑)       | FID (↓)        | Accuracy (↑)           | Forgetting (↓)  |
> ||
> | MoELoRA [5]       | 0.2094               | **0.5234**                | 63.26          | **34.14**              | **24.08**       |
> | MoELoRA+KD     | **0.2568**           | 0.5176                    | **52.81**      | 32.41                  | 25.27           |
> ||
> | CL‑MoE [6]        | 0.2370               | 0.5146                    | 54.69          | **36.96**              | **19.12**       |
> | CL‑MoE+KD      | **0.2439**           | **0.5200**                | **52.43**      | 30.68                  | 21.23           |
>
> **2. Decoupled system without KD.**
> As shown in Table R3, decoupling textual and visual LoRA without knowledge distillation yields only modest gains in understanding accuracy and visual generation quality compared to vanilla LoRA. This indicates that decoupling alone is insufficient to maintain the rich cross-modal alignment necessary for high-quality generation. Furthermore, Section 5.3 of the main paper includes an ablation on “MoDE w/o KD” (i.e., decoupled T-MoE and V-adapter without knowledge distillation), which further confirms that knowledge distillation plays a critical role in the performance improvements observed.
>
> **Table R3: Ablation on Fully De-coupled Architecture.**
>
> | Method           || **Image Generation** || **Visual Understanding** ||
> |------------------|:----------------------:|:----------------------------:|:----------------:|:------------------------:|:-----------------:|
> |                  | Text alignment (↑)   | Image alignment (↑)       | FID (↓)        | Accuracy (↑)           | Forgetting (↓)  |
> ||
> | SeqLoRA          | 0.2183               | 0.5150                    | 56.75          | 28.75                  | 26.83           |
> | Decoupled LoRA   | 0.2252               | 0.5180                    | 54.74          | 29.07                  | 27.60           |
>
> The above ablations show that MoDE is more than the sum of its parts: only the combination of modality‑decoupling plus our specialized distillation leads to consistently superior trade‑offs.
>
> ## Weakness 4: Concurrent works
> Thank you for the valuable suggestion to discuss concurrent works on better unified modeling and to clarify whether our contributions are orthogonal to these approaches. Both BAGEL [2] and OmniMamba [3] adopt decoupled encoders, and OmniMamba further applies decoupled LoRA adapters, which have similarities to our approach. This indicates that the design of modality-decoupled experts has been recognized as an effective solution. However, their focus is primarily on architectural decoupling for pretraining or static tasks, rather than continual instruction tuning. In contrast, our work addresses the challenges of continual adaptation, which is essential for real-world scenarios such as lifelong multimodal assistants and adaptive AI systems that learn and evolve over time.
>
> ## Q1: KD on top of seq LoRA
> Thank you for the question regarding the effect of applying knowledge distillation (KD) on top of sequential LoRA.
> We conducted an ablation to evaluate the effect of adding KD on top of vanilla sequential LoRA fine-tuning. As shown in Table R4 below, applying KD leads to modest improvements in both image generation quality (higher alignment scores, lower FID) and multimodal understanding performance (higher accuracy, reduced forgetting).
> This suggests that KD alone can partially mitigate forgetting and preserve visual generation quality. However, as shown in other results, KD without proper modality decoupling may still interfere with understanding tasks. MoDE outperforms SeqLoRA+KD by explicitly addressing this interference through modality separation.
>
> **Table R4: Effect of knowledge distillation on SeqLoRA.**
>
> | Method           || **Image Generation** || **Visual Understanding** ||
> |------------------|:----------------------:|:----------------------------:|:----------------:|:------------------------:|:-----------------:|
> |                  | Text alignment (↑)   | Image alignment (↑)       | FID (↓)        | Accuracy (↑)           | Forgetting (↓)  |
> ||
> | SeqLoRA          | 0.2183               | 0.5150                    | 56.75          | 28.75                  | 26.83           |
> | SeqLoRA + KD | 0.2531           | 0.5176                | 52.22      | 30.54              | 23.18       |
> ### References
> [1]: Schuhmann, Christoph, et al. "Laion-5b: An open large-scale dataset for training next generation image-text models." Advances in neural information processing systems 35 (2022): 25278-25294.
>
> [2]: Deng, Chaorui, et al. "Emerging properties in unified multimodal pretraining." arXiv preprint arXiv:2505.14683 (2025).
>
> [3]: Zou, Jialv, et al. "Omnimamba: Efficient and unified multimodal understanding and generation via state space models." arXiv preprint arXiv:2503.08686 (2025).
>
> [4]: Xu, Zhiyang, et al. "Modality-Specialized Synergizers for Interleaved Vision-Language Generalists." The Thirteenth International Conference on Learning Representations. 2025.
>
> [5]: Chen, Cheng, et al. "Coin: A benchmark of continual instruction tuning for multimodel large language models." Advances in Neural Information Processing Systems 37 (2024): 57817-57840.
>
> [6]: Huai, Tianyu, et al. "CL-MoE: Enhancing Multimodal Large Language Model with Dual Momentum Mixture-of-Experts for Continual Visual Question Answering." Proceedings of the Computer Vision and Pattern Recognition Conference. 2025.

---

### Official Review · Reviewer_LAT9 · 2025-06-11

**Clarity:** 3
**Significance:** 3
**Originality:** 2
**Rating:** 5
**Confidence:** 3

**Summary:**

In this work, the authors propose Modality-Decoupled Experts (MoDE). This architecture handles intra- and inter-modal forgetting for Unified Multimodal Generative Models (UMGMs). MoDE consists of two modules: a V-adapter (a LoRA module) used to capture visual knowledge and a T-MoE, a LoRA Mixture of Experts that handles textual information. By decoupling the vision and language modalities, the authors claim to reduce forgetting when training the model on a sequence of tasks. The authors provide a wide range of experiments to validate their thesis.

**Questions:**

1. How does MoDE compare to CL-MoE regarding time and memory costs?

2. The reference dataset is not clear. I understand it is used for the KD loss, but how is it composed? Is it task-related, or is it always the same?

Please refer also to the weaknesses.

**Ethical Concerns:**

["NO or VERY MINOR ethics concerns only"]

**Final Justification:**

Authors have addressed all concerns, thus I increase my rating to Accept.

**Limitations:**

The authors have not discussed limitations. One may be the hardware requirements.

**Quality:**

3

**Strengths And Weaknesses:**

### Strengths

- The paper is well-written and easy to follow.

- The problem of dealing with forgetting in both modalities of a UMGM is interesting and worth exploring.

- The experiments section and appendix are exhaustive

### Weaknesses

- While the results are consistently better than previous approaches, they are incremental with respect to CL-MoE (+0.1 in Tab. 2). This may raise some questions about the approach's statistical validity. Multiple seeded runs would have helped to understand this aspect better.

- The V-Adapter and T-MoE have never really been introduced. They are explained only in the caption of Fig. 3.

- Details and ablations on the number of experts in T-MoE are missing.

---

> ### Author Rebuttal · Authors · 2025-07-31
>
> We thank Reviewer LAT9 for the positive feedback on the clarity of our writing, the importance of addressing forgetting in both modalities of UMGMs, and the thoroughness of our experiments and appendix. We appreciate the constructive comments and address the raised concerns in detail below.
> ## Weakness 1: Statistical validity of results in Tab. 2
> We thank the reviewer for raising this important point of the statistical significance of the improvements over CL-MoE.
> To evaluate the robustness of our results, we conducted 3 runs with different random seeds for both CL-MoE and MoDE. The results are summarized in Table R1 below. We report the mean and standard deviation across 3 runs.
>
> **Table R1: Results with multiple seeds (with mean and standard deviation). For each method, the
> upper row shows the best accuracy during continual instruction tuning, and the lower row shows the
> final accuracy after completing all tasks.**
>
> | Method         | ScienceQA            | TextVQA              | ImageNet            | GQA                  | VizWiz              | Grounding           | ACC (↑)      | Fgt (↓)      |
> |----------------|----------------------|-----------------------|----------------------|-----------------------|----------------------|----------------------|--------------|--------------|
> | CL‑MoE [2]        | 71.35±0.01           | 38.88±0.06            | 90.12±0.04           | 37.37±0.04            | 43.74±0.07           | 35.76±0.07           |              |              |
> |                | 67.32±0.01           | 32.82±0.01            | 9.13±0.04            | 33.30±0.04            | 43.20±0.01           | 35.76±0.07           | 36.92    | 19.14    |
> ||
> | **MoDE (Ours)**| 71.49±0.01           | 39.00±0.01            | 85.88±0.04           | 36.73±0.01            | 44.45±0.01           | 38.55±0.08           |              |              |
> |                | 67.87±0.01           | 33.36±0.01            | 9.54±0.01            | 33.60±0.01            | 41.91±0.01           | 38.55±0.08           | **37.47**    | **18.25**    |
>
> Although the absolute improvement in ACC may appear small (e.g., +0.55), MoDE consistently improves both accuracy and forgetting across all seeds and datasets. Given the small standard deviations, the performance gains are meaningful and robust. These results strengthen the statistical significance and reliability of our conclusions.
>
> ## Weakness 2: Introduction of V-Adapter and T-MoE
> Thank you for pointing out that V-Adapter and T-MoE are not formally introduced in the main text and are only briefly mentioned in the caption of Figure 3. We agree that the current version does not clearly introduce V-Adapter and T-MoE in the main text. In the revised version, we will explicitly define them in Section 4.2 Modality-Decoupled Experts (MoDE).
>
> ## Weakness 3: Ablation on the number of experts in T-MoE
> Thank you for the suggestion. We provide an ablation study on the number of experts in the T-MoE module. As shown in Table R2 below, increasing the number of experts from 4 to 6 yields slightly improved text alignment and accuracy, though the gains are modest. This suggests that the performance of MoDE is relatively robust to the number of experts, and 4 experts offer a good trade-off between performance and computational cost.
>
> **Table R2: Ablation on the number of experts in T-MoE.**
>
> | #Experts || **Image Generation** || **Visual Understanding** ||
> |----------|:----------------------:|:----------------------------:|---------:|:------------------------:|:-----------------:|
> |          | Text alignment (↑)   | Image alignment (↑)        | FID (↓) | Accuracy (↑)           | Forgetting (↓)  |
> ||
> | 4        | 0.2587               | 0.5190                     | 51.27   | 37.05                  | 18.25           |
> | 6        | 0.2592               | 0.5166                     | 51.13   | 37.26                  | 18.64           |
>
> ## Q1: Time and Memory Costs Compared to CL-MoE
> Thank you for the question of "How does MoDE compare to CL-MoE regarding time and memory costs?".
> We compare MoDE with CL-MoE and MoELoRA in terms of TFLOPs, peak GPU memory, training time per iteration, and the percentage of trainable parameters. Results are shown in Table R3 below.
> MoDE incurs a slightly higher computational cost and memory usage than CL-MoE due to its use of both modality-specific adapters and knowledge distillation. However, the differences are modest and well within practical bounds. In return, MoDE consistently yields better retention and overall performance, demonstrating a favorable trade-off between efficiency and effectiveness.
>
> **Table R3: Comparison of training efficiency and memory usage.**
>
> | Method        | TFLOPs | Peak GPU Memory (MB) | Training Time (ms) | Trainable Params (%) |
> |---------------|--------|----------------------|---------------------|-----------------------|
> | MoELoRA [3]      | 4.29   | 75265               | 419.2               | 0.0171                |
> | CL‑MoE [2]       | 4.29   | 86300               | 421.7               | 0.0171                |
> | **MoDE (Ours)**   | 4.19   | 84980               | 440.3               | 0.0211                |
>
> ## Q2: Reference dataset
> Thank you for the question regarding the composition of the reference dataset used for knowledge distillation (KD). The reference dataset used for KD loss is composed of text prompts sampled from the LAION-5B dataset [1]. The dataset is shared across all tasks, rather than being tailored to the current task. The shared reference dataset ensures that the KD signal reflects general visual generation capability, independent of the downstream multimodal understanding task. We will clarify this more explicitly in the revised manuscript.
> ### References
> [1]: Schuhmann, Christoph, et al. "Laion-5b: An open large-scale dataset for training next generation image-text models." Advances in neural information processing systems 35 (2022): 25278-25294.
>
> [2]: Huai, Tianyu, et al. "CL-MoE: Enhancing Multimodal Large Language Model with Dual Momentum Mixture-of-Experts for Continual Visual Question Answering." Proceedings of the Computer Vision and Pattern Recognition Conference. 2025.
>
> [3]: Chen, Cheng, et al. "Coin: A benchmark of continual instruction tuning for multimodel large language models." Advances in Neural Information Processing Systems 37 (2024): 57817-57840.

---

> > ### Comment · Reviewer_LAT9 · 2025-08-04
> >
> > Thank you for your response, I will update my rating to Accept.

---

> > > ### Author Response · Authors · 2025-08-04
> > >
> > > Thank you very much for your positive feedback. We sincerely appreciate your updated evaluation and recognition of our work.

---

### Official Review · Reviewer_qoBP · 2025-06-17

**Clarity:** 3
**Significance:** 3
**Originality:** 3
**Rating:** 4
**Confidence:** 4

**Summary:**

This paper is the first to investigate the inter-modal catastrophic forgetting issue in Unified Multimodal Understanding and Generation Models (UMGMs) during continual instruction tuning. While previous work on multimodal continual instruction tuning has focused solely on understanding tasks, this paper explores how UMGMs forget their visual generation capabilities after being continually fine-tuned on multimodal understanding tasks. The authors attribute this forgetting to gradient conflicts between image generation loss and text generation loss with respect to shared model parameters and provide a theoretical explanation for this phenomenon. To mitigate this problem, the authors propose a modality-decoupled expert architecture. Specifically, they introduce a MoELoRA module for text token generation (referred to as T-MoE), and a separate LoRA module for image token generation (referred to as V-Adapter). To preserve the image generation ability, they further apply knowledge distillation from the original (teacher) UMGM to the V-Adapter in the new (student) model.

**Questions:**

- The MoELoRA (CoIN) benchmark includes 8 multimodal understanding tasks: ScienceQA, TextVQA, ImageNet, GQA, VizWiz, Grounding, VQAv2, and OCR-VQA. Why does the evaluation in this paper only cover the first six tasks? What are the results of all 8 tasks?

- In the supplementary material, the results for DualPrompt and MoELoRA are missing in the different task ordering section. Also, the Janus-Pro results are missing for DualPrompt, Model Tailor, and MoELoRA. Could these be included?

- According to Table 2, MoDE only shows marginal improvements in multimodal understanding performance (both in the best accuracy during continual instruction tuning and the final accuracy after completing all tasks) compared to MoELoRA and CL-MoE. The main benefit seems to lie in mitigating the forgetting in image generation. If knowledge distillation for image generation were directly applied to MoELoRA and CL-MoE, how would they perform? Additionally, can the authors report the detailed accuracy on previous tasks after each task training step?

- In the current tables, bold fonts highlight the results of the proposed method. It might be clearer to instead bold the best results for each metric.

**Ethical Concerns:**

["NO or VERY MINOR ethics concerns only"]

**Final Justification:**

The author's response has addressed my concerns.

**Limitations:**

Yes.

**Paper Formatting Concerns:**

No.

**Quality:**

3

**Strengths And Weaknesses:**

Pros:

- The paper is the first to investigate the catastrophic forgetting of image generation capabilities in UMGMs under continual instruction tuning.

- A theoretical explanation is provided by analyzing the gradient conflicts between image token generation loss and text token generation loss with respect to shared model parameters.


Cons:

- The paper assumes that image generation is only conducted during pretraining and that continual instruction tuning only involves multimodal understanding tasks. If the model is further fine-tuned on image generation tasks, will it then catastrophically forget the understanding tasks? Can the proposed method prevent this problem as well?

- The “Reference Token” in Figure 3 is not clearly explained. When knowledge distillation is applied to image generation during continual tuning, what text prompt is used as the reference?

---

> ### Author Rebuttal · Authors · 2025-07-31
>
> We thank Reviewer qoBP for highlighting the novelty of our work as the first to study catastrophic forgetting of image generation in UMGMs under continual instruction tuning, and for appreciating our theoretical explanation based on gradient conflicts between image and text generation losses. The questions raised are insightful, and we provide our detailed responses below.
>
> ## Weakness 1: Fine-tuning on image generation tasks
> Thank you for the insightful question: whether fine-tuning on image generation tasks would lead to catastrophic forgetting of the understanding capabilities, and whether our proposed method can mitigate such forgetting.
> To address this, we conduct new experiments by fine-tuning the linear layers of the Chameleon model [3] on image generation tasks. Specifically, we use the LAION-5B dataset [1] for fine-tuning and apply knowledge distillation using the multimodal understanding sequence from the GQA dataset [2] as reference data. We compare our MoDE method with LoRA fine-tuning across three multimodal understanding benchmarks. Accuracies are reported in Table R1. The difference between zero-shot and fine-tuned performance reflects the degree of forgetting in multimodal understanding.
>
> **Table R1: Forgetting caused by fine-tuning on image generation tasks.**
>
> | Method        | ScienceQA | TextVQA | ImageNet |
> |---------------|-----------|---------|----------|
> | Zero-shot     | 51.52     | 23.49   | 16.53    |
> | LoRA          | 45.81     | 12.25   | 0.00     |
> | **MoDE (Ours)**   | 51.28     | 23.44   | 16.61    |
>
> These results support two conclusions:
>
> (1) Yes, fine-tuning on image generation tasks can indeed cause catastrophic forgetting in multimodal understanding, confirming that the forgetting problem is bidirectional. This is evident in the substantial drop in performance for the LoRA baseline compared to zero-shot.
>
> (2) Our MoDE method remains effective in this scenario, significantly reducing the performance drop and mitigating forgetting relative to the LoRA baseline. This demonstrates that MoDE generalizes beyond the specific forgetting direction explored in the main paper, where understanding tasks overwrite prior image generation capabilities, and is effective regardless of the fine-tuning direction.
>
> ## Weakness 2: "Reference Data" in Figure 3
> In our experiments, "Reference Data" are text prompts selected from LAION-5B dataset [1].
>
> ## Q1: CoIN benchmark
> Thank you for pointing this out. Due to resource constraints, we selected a subset of 6 tasks from the full CoIN benchmark to ensure feasibility while maintaining diversity across modalities and task types. Specifically, the chosen tasks (ScienceQA, TextVQA, ImageNet, GQA, VizWiz, and Grounding) span a representative range of vision-language reasoning, recognition, and grounding challenges.
> While we did not include VQAv2 and OCR-VQA in our current evaluation, we expect similar trends to hold, given the consistency of results across the six evaluated tasks. Extending evaluation to all 8 tasks is a valuable contribution and we plan to include it in the final paper.
>
> ## Q2: Additional supplementary results
> Thank you for the helpful suggestion regarding the missing results for DualPrompt and MoELoRA under different task orderings, as well as the missing Janus-Pro results for several baselines in the supplementary material. We provide the complete results below:
>
> **Q2.1:** We report additional ablations on task order sensitivity in Table R2. DualPrompt [5] and MoELoRA [4] are now included alongside MoDE. DualPrompt yields low accuracy and low forgetting. This is because adding a small set of prompt embeddings slightly biases the feature representations, which is insufficient to learn the new and hard tasks. The results in Table R2 show that MoDE maintains superior performance and robustness under alternate task orders.
>
> **Table R2: Additional ablation on task order.**
>
> | Method           ||**Image Generation**|| **Visual Understanding** ||
> |------------------|:----------------------:|:----------------------------:|:----------------:|:------------------------:|:-----------------:|
> |                  | Text alignment (↑)   | Image alignment (↑)       | FID (↓)        | Accuracy (↑)           | Forgetting (↓)  |
> ||
> | DualPrompt [5] | 0.2491               | 0.5140                    | 53.07          | 27.43                  | **7.09**         |
> | MoELoRA [4] | 0.2492               | 0.5125                    | 61.64          | 33.37                  | 23.61           |
> | **MoDE (Ours)**  | **0.2558**           | **0.5256**                | **51.69**      | **35.11**              | 18.25           |
>
>
> **Q2.2:** In Table R3, we provide the missing Janus-Pro results for Model Tailor [6], MoELoRA[4], and our MoDE. Again, MoDE demonstrates the best performance with the lowest forgetting and highest accuracy.
>
> **Table R3: Evaluation on Janus-Pro benchmark.**
>
> | Method           | Accuracy (↑) | Forgetting (↓) |
> |------------------|--------------|----------------|
> | Model Tailor [6] | 29.13        | 32.01          |
> | MoELoRA [4] | 24.61        | 36.10          |
> | **MoDE (Ours)**  | **53.02**    | **14.48**      |
>
> ## Q3: KD on MoELoRA and CL-MoE
> Thank you for the thoughtful question about the effect of applying image generation knowledge distillation to MoELoRA and CL-MoE, and whether MoDE’s benefit extends beyond that.
> To directly evaluate the effect of applying knowledge distillation (KD) for image generation on existing baselines, we conduct new experiments by incorporating KD into MoELoRA and CL-MoE. The results are shown in Table R4.
>
> **Table R4: Effect of KD on MoELoRA and CL-MoE.**
>
> | Method           || **Image Generation** || **Visual Understanding** ||
> |------------------|:----------------------:|:----------------------------:|:----------------:|:------------------------:|:-----------------:|
> |                  | Text alignment (↑)   | Image alignment (↑)       | FID (↓)        | Accuracy (↑)           | Forgetting (↓)  |
> ||
> | MoELoRA [4] | 0.2094 | 0.5234 | 63.26 | 34.14 | 24.08 |
> | MoELoRA + KD | 0.2568 | 0.5176 | 52.81 | 32.41 | 25.27 |
> ||
> | CL‑MoE [7] | 0.2370 | 0.5146 | 54.69 | 36.96 | 19.12 |
> | CL‑MoE + KD | 0.2439 | **0.5200** | 52.43 | 30.68 | 21.23 |
> ||
> | **MoDE (Ours)** | **0.2587** | **0.5200** | **51.27** | **37.05** | **18.25** |
>
> These results show that applying KD improves image generation quality for both MoELoRA and CL-MoE, as indicated by higher alignment scores and lower FIDs. However, KD degrades the multimodal understanding performance (reducing accuracy and increasing forgetting) because the models are not decoupled, and the KD interferes with learning on understanding tasks. In contrast, MoDE achieves the best performance across all metrics by design: its modality-decoupled structure prevents interference from KD and enables effective retention of multimodal understanding.
>
> ## Q4: Bold fonts
> Thanks for valuable suggestion and we will bold the best results for every metric.
> ### References
> [1]: Schuhmann, Christoph, et al. "Laion-5b: An open large-scale dataset for training next generation image-text models." Advances in neural information processing systems 35 (2022): 25278-25294.
>
> [2]: Hudson, Drew A., and Christopher D. Manning. "Gqa: A new dataset for real-world visual reasoning and compositional question answering." Proceedings of the IEEE/CVF conference on computer vision and pattern recognition. 2019.
>
> [3]: Team, Chameleon. "Chameleon: Mixed-modal early-fusion foundation models." arXiv preprint arXiv:2405.09818 (2024).
>
> [4]: Chen, Cheng, et al. "Coin: A benchmark of continual instruction tuning for multimodel large language models." Advances in Neural Information Processing Systems 37 (2024): 57817-57840.
>
> [5]: Wang, Zifeng, et al. "Dualprompt: Complementary prompting for rehearsal-free continual learning." European conference on computer vision. Cham: Springer Nature Switzerland, 2022.
>
> [6]: Zhu, Didi, et al. "Model tailor: Mitigating catastrophic forgetting in multi-modal large language models." arXiv preprint arXiv:2402.12048 (2024).
>
> [7]: Huai, Tianyu, et al. "CL-MoE: Enhancing Multimodal Large Language Model with Dual Momentum Mixture-of-Experts for Continual Visual Question Answering." Proceedings of the Computer Vision and Pattern Recognition Conference. 2025.

---

> > ### Comment · Reviewer_qoBP · 2025-08-04
> >
> > Thanks for your response. I have updated my rating.

---

> > > ### Author Response · Authors · 2025-08-04
> > >
> > > Thank you very much for your updated rating and for your thoughtful feedback during the review process.

---

### Official Review · Reviewer_vBLd · 2025-07-02

**Clarity:** 3
**Significance:** 2
**Originality:** 2
**Rating:** 5
**Confidence:** 5

**Summary:**

This paper explores the challenge of catastrophic forgetting in Unified Multimodal Generative Models (UMGMs), focusing particularly on the phenomenon of inter-modal forgetting, alongside intra-modal forgetting. The paper identifies gradient conflict between visual and textual modalities as a key source of inter-modal forgetting and provides several theoretical contributions showing that such gradient conflicts directly cause inter-modal forgetting. The authors propose Modality-Decoupled Experts (MoDE), a straightforward approach that decouples visual and textual modality updates using modality-specific LoRA adapters, along with knowledge distillation from a frozen visual generation teacher to prevent intra-modal forgetting. MoDE outperforms competitive baselines such as MoE-LoRA and Model-Tailer on a number of multimodal continual learning tasks, and significantly reduces both intra- and inter-modal forgetting.

**Questions:**

1. A comparison of the trainable parameter and computational costs between the proposed method and the baselines would help alleviate concerns of triviality of the proposed method.
2. An ablation showing a trivial fully de-coupled system with separate image-generation and text/multimodal understanding modules, would be helpful.
3. Does Figure 1 training use LoRA adapters to measure forgetting? Why was forgetting caused by full fine-tuning not evaluated?

_Points 1 and 2 above, as well as a clarification of the first 3 points expressed in the Weaknesses section, would go farthest in helping alleviate concerns._

**Ethical Concerns:**

["NO or VERY MINOR ethics concerns only"]

**Final Justification:**

I thank the authors for an excellent response to the numerous queries and concerns. In particular, the ablations showing decoupling without KD, and adding KD without decoupling both perform materially worse is a strong support to the central arguments of the paper. In combination with the strong results, I believe it is easy to recommend acceptance for this paper.

I nevertheless encourage the authors to clarify their terminologies. In particular, please ensure the focus on only visual forgetting as inter-modal forgetting is clarified early in the paper, and that text-only forgetting is excluded from the study is clarified in the related work.

**Limitations:**

Yes, Addressed.

**Paper Formatting Concerns:**

None of note.

**Quality:**

3

**Strengths And Weaknesses:**

**Strengths:**
- **Novel Theoretical Contributions:** The paper makes a number of theoretical formulations which are interesting and broadly useful:
    - The proposed formulation of the Modality Gradient conflict, showing that an increase in image generation loss (forgetting) depends on the inner product of the formulated visual and text gradients is interesting. The formulation also shows how the proposed MoDE method provably bounds inter-modal interference to quadratic in step sizes.
    - These are generally useful formulations, and would be of interest to future work in mitigating catastrophic forgetting in Unified Multimodal Generative Models (UMGMs).

- **Strong results in continual instruction tuning \& mitigation of catastrophic forgetting:**
    - In both mitigation of catastrophic forgetting in image generation and visual understanding, as well as in continual instruction tuning experiments the proposed method MoDE almost consistently outperforms strong baselines like Model-tailor and MoE-LoRA.
    - While this may be due to the trivial modality decoupling in the proposed MoDE, the approach achieves lower forgetting (and higher final accuracy) on almost all tasks.

- **Well motivated study:**
  - The unification and study of intra-model and inter-modal forgetting is interesting and well motivated.

**Weaknessess:**

- **Core method seems to lack novelty and significance**: The core mechanisms in the proposed MoDE method leverage existing ideas in modality-specific and MoE adapters.
  - To prevent inter-modal forgetting, the proposed method involves de-coupling both visual and textual modalities by learning separate modules - Mixture-of-Expert LoRAs for Text and a Visual LoRA for vision. This seems like a trivial solution, and re-uses existing modules (MoE-LoRA, LoRA for vision adaptation).
  - In addition, to prevent intra-modal forgetting (the authors only consider visual modality forgetting), MoDE uses a straightforward knowledge distillation with a frozen pre-trained teacher backbone.


- **Strong results in large part due to modality decoupling, Missing decoupled baselines:**
    - While the results of the proposed approach themselves are strong, it seems this is largely a result of the trivial modality decoupling used in the proposed approach.
    - There is no clear baseline comparing the proposed approach to a trivial modality decoupled system with separate image-generation and text/multimodal understanding modules. It is therefore difficult to gauge the usefulness of the proposed approach.


- **Missing Parameter and Computational costs comparisons:**
    - Since the proposed MoDE essentially leverages separate LoRA adapters for both modalities as well as for different multimodal understanding tasks, it would be interesting to see a comparison of the trained parameters and the training computational costs between the proposed method and baselines.

- **Text-only intra-modal forgetting:** Inter-modal forgetting is missing forgetting caused to text-only generation tasks, while only visual generation has been considered. This seems like a critical aspect of inter-modal forgetting. See [1,2]:

[1]: Zhang, Yi-Kai, et al. "Wings: Learning multimodal llms without text-only forgetting." Advances in Neural Information Processing Systems 37 (2024): 31828-31853.

[2]: Srivastava, Shikhar, et al. "Improving multimodal large language models using continual learning." arXiv preprint arXiv:2410.19925 (2024).

- **Minor corrections:**
   -  VQA tasks are repeatedly referred to as text-only (Line 122-123, for instance) in the paper. While the generation is in text, the task however is _not_ "text-only" since these are VQA.  This is very odd, and must be clarified/corrected in the paper.
    - Line 122-123, Line 50: "(e.g., improving language generation while preserving visual generation)" where this isn't language generation, it's multimodal understanding.

---

> ### Author Rebuttal · Authors · 2025-07-31
>
> We thank Reviewer vBLd for the positive assessment of our paper’s theoretical contributions, strong empirical results in continual instruction tuning, and the well-motivated study of intra- and inter-modal forgetting in unified multimodal models. The questions raised are insightful, and we will address each point below to clarify our contributions and the specifics of our work.
>
> ## Weaknesses 1&2, Q2: Novelty and decoupled baselines
> We understand the concern that our building blocks (MoE‑LoRA, LoRA, KD) have appeared in prior work. To demonstrate that their combination in MoDE is neither trivial nor interchangeable, we conducted the following ablations:
>
> **1. KD without decoupling.**
> As shown in Table R1, adding our cross‑modal KD loss to CL‑MoE or to MoELoRA reduces multimodal understanding accuracy (e.g. 34.14% to 32.41% for MoELoRA), even though it does mitigate visual‑generation forgetting. This is because without explicit modality decoupling, KD “locks in” the teacher’s features in a way that hinders the model’s ability to adapt to new VQA and grounding tasks.
>
> **Table R1: Effect of adding KD to MoELoRA and CL-MoE.**
>
> | Method         ||**Image Generation** || **Visual Understanding** ||
> |----------------|:----------------------:|:----------------------------:|:----------------:|:------------------------:|:-----------------:|
> |                | Text alignment (↑)   | Image alignment (↑)       | FID (↓)        | Accuracy (↑)           | Forgetting (↓)  |
> ||
> | MoELoRA [2] | 0.2094               | **0.5234**                | 63.26          | **34.14**              | **24.08**       |
> | MoELoRA+KD     | **0.2568**           | 0.5176                    | **52.81**      | 32.41                  | 25.27           |
> ||
> | CL‑MoE [3]        | 0.2370               | 0.5146                    | 54.69          | **36.96**              | **19.12**       |
> | CL‑MoE+KD      | **0.2439**           | **0.5200**                | **52.43**      | 30.68                  | 21.23           |
>
> **2. Decoupled system without KD.**
> As shown in Table R2, decoupling textual and visual LoRA without knowledge distillation yields only modest gains in understanding accuracy and visual generation quality compared to vanilla LoRA. This indicates that decoupling alone is insufficient to maintain the rich cross-modal alignment necessary for high-quality generation. Furthermore, Section 5.3 of the main paper includes an ablation on “MoDE w/o KD” (i.e., decoupled T-MoE and V-adapter without knowledge distillation), which further confirms that knowledge distillation plays a critical role in the performance improvements observed.
>
> **Table R2: Ablation on Fully De-coupled Architecture.**
>
> | Method           ||**Image Generation**|| **Visual Understanding** ||
> |------------------|:----------------------:|:----------------------------:|:----------------:|:------------------------:|:-----------------:|
> |                  | Text alignment (↑)   | Image alignment (↑)       | FID (↓)        | Accuracy (↑)           | Forgetting (↓)  |
> ||
> | SeqLoRA          | 0.2183               | 0.5150                    | 56.75          | 28.75                  | 26.83           |
> | Decoupled LoRA   | 0.2252               | 0.5180                    | 54.74          | 29.07                  | 27.60           |
>
> In conclusion, the above ablations show that MoDE is more than the sum of its parts: only the combination of modality‑decoupling plus knowledge distillation leads to consistently superior performance.
> ## Weakness 3 & Q1: Parameter and computational costs comparisons
> We appreciate the reviewer’s suggestion. To address concerns regarding the triviality of our method, we report a comparison of training compute, memory footprint, and trainable parameters in the table below. As shown, MoDE introduces only a slight increase in training time (440.3 ms vs. 419.2 ms) and peak GPU memory (84.98 GB vs. 75.27 GB) compared to MoELoRA, while maintaining comparable TFLOPs. The trainable parameter ratio of MoDE is only 0.0211%, modestly higher than MoELoRA and CL-MoE (0.0171%), indicating that our performance gains are not simply due to increased processing capacity. These results suggest that MoDE's improvements arise from its design (modality-decoupled experts and distillation), rather than significantly higher resource usage.
>
> **Table R3: Comparison of Trainable Parameters and Training Costs.**
>
> | Method        | TFLOPs | Peak GPU Memory (MB) | Training Time (ms) | Trainable Params (%) |
> |---------------|:--------:|:----------------------:|:---------------------:|:-----------------------:|
> | MoELoRA [2] | 4.29 | 75265 | 419.2 | 0.0171 |
> | CL‑MoE [3] | 4.29 | 86300 | 421.7 | 0.0171 |
> | **MoDE (Ours)** | 4.19 | 84980 | 440.3 | 0.0211 |
>
> ## Weakness 4: Text-only inter-modal forgetting
> We appreciate the reviewer’s insightful comment and references. This is the first work on inter-modal forgetting on UMGMs continual learning, so we use the forgetting in visual generation caused by continual instruction tuning on multimodal understanding as the study case. This is a very interesting direction; we will explore it in our future work. However, this is currently beyond the scope of our study, which aims to establish the foundational analysis and mitigation strategies for inter-modal interference through the lens of visual generation degradation.
>
> ## Weakness 5: Correction of terminology
> We apologize for the confusion caused by the misuse of "text-only generation" and "language generation" and will replace them with "multimodal understanding" in the revised paper.
>
> ## Q3: Figure 1 explanation
> The question raised concerns about whether Figure 1 uses LoRA adapters to measure forgetting, and why forgetting caused by full-model fine-tuning was not evaluated.
> Yes, Figure 1 is obtained using LoRA-based fine-tuning. While we acknowledge that full-model fine-tuning may lead to different absolute values, we chose LoRA due to practical constraints on compute resources and because it has become a widely used paradigm for efficient continual adaptation in large-scale models.
> Importantly, our goal is to study the modality imbalance and forgetting phenomena under realistic fine-tuning setups. LoRA provides a representative and resource-efficient way to examine these effects in current practice. Moreover, prior work (e.g., [1]) has shown that full fine-tuning tends to exacerbate catastrophic forgetting in MLLMs, so our reported forgetting levels are likely conservative.
> Therefore, even under LoRA, the modality-specific forgetting patterns are clearly observable and support the motivation and effectiveness of our proposed MoDE approach.
> ### References
> [1]: Zhu, Didi, et al. "Model tailor: Mitigating catastrophic forgetting in multi-modal large language models." arXiv preprint arXiv:2402.12048 (2024).
>
> [2]: Chen, Cheng, et al. "Coin: A benchmark of continual instruction tuning for multimodel large language models." Advances in Neural Information Processing Systems 37 (2024): 57817-57840.
>
> [3]: Huai, Tianyu, et al. "CL-MoE: Enhancing Multimodal Large Language Model with Dual Momentum Mixture-of-Experts for Continual Visual Question Answering." Proceedings of the Computer Vision and Pattern Recognition Conference. 2025.

---

> > ### Comment · Reviewer_vBLd · 2025-08-05
> >
> > I thank the authors for an excellent response to the numerous queries and concerns.
> > In particular, the ablations showing decoupling without KD, and adding KD without decoupling both perform materially worse is a strong support to the central arguments of the paper.  In combination with the strong results, I believe it is reasonable to recommend acceptance for this paper.
> >
> > I nevertheless encourage the authors to clarify their terminologies. In particular, please ensure the focus on only visual forgetting as inter-modal forgetting is clarified early in the paper, and that text-only forgetting is excluded from the study is clarified in the related work.

---

> > > ### Author Response · Authors · 2025-08-05
> > >
> > > Thank you very much for the encouraging feedback and for your willingness to support our paper’s acceptance. We truly appreciate your recognition of the ablation studies and how they strengthen the central arguments of our work.
> > >
> > > We will certainly incorporate your suggestions in the camera-ready version. Specifically, we will clarify the use of our terminologies early in the paper, emphasize our focus on visual forgetting as a form of inter-modal forgetting, and explicitly note in the related work section that text-only forgetting is excluded from our study.

---

### Note · Authors · 2025-08-13

We sincerely thank the reviewers and AC for their constructive feedback, and recognition of our paper’s contributions in (i) identifying and theoretically analyzing inter-modal forgetting in Unified Multimodal Generative Models (UMGMs), and (ii) proposing MoDE, which effectively mitigates both intra- and inter-modal forgetting.

In the rebuttal, we provided new experiments and ablations to directly address the main concerns:

**Novelty vs. triviality** – Our ablations show that neither modality decoupling nor knowledge distillation (KD) alone can match MoDE’s trade-offs. KD without decoupling degrades multimodal understanding, while decoupling without KD fails to preserve rich cross-modal alignment. Only their joint integration in MoDE yields consistent gains.

**Decoupled baselines and KD on existing methods** – Our experiments confirm MoDE’s advantage over “fully decoupled” LoRA and over MoELoRA/CL-MoE augmented with KD, thus demonstrating the benefits of our design.

**Compute and memory cost** – MoDE achieves its improvements with negligible overhead in TFLOPs, memory, and trainable parameters compared to strong baselines (e.g., CL-MoE, MoELoRA).

**Bidirectional forgetting** – Our new results show that MoDE also mitigates forgetting of multimodal understanding when fine-tuned on image generation, indicating generality beyond the main paper’s focus.

**Statistical robustness** – Multi-seed runs confirm consistent accuracy and forgetting improvements over CL-MoE with low variance.

**Clarifications** – We will clearly define V-Adapter/T-MoE in the main text, correct “text-only” terminology, and state early that our inter-modal forgetting study focuses on visual generation degradation. The LAION-5B reference set used for KD will be described explicitly.

We believe these additions strengthen both the methodological soundness and empirical validity of our work, and address all substantive reviewer concerns. We appreciate the reviewers’ updated ratings and support for acceptance, and look forward to incorporating these clarifications in the camera-ready version.

---

### Decision · Program_Chairs · 2025-09-17

**Decision:**

Accept (poster)

**Comment:**

The paper introduces inter-modal forgetting in unified multimodal generative models, formalizes its cause through gradient conflict analysis, and proposes a solution. Experiments show how MoDE preserves image generation quality while sustaining multimodal understanding accuracy, outperforming baselines.

Strengths: The paper identifies and formalizes an important and timely problem, and presents an effective mitigation.

Weaknesses: The mechanism (decoupling + KD) is incremental, with novelty primarily in problem framing. Evaluation is limited to one main task order and a 6-task stream (rather than the full 8 task stream, without much justification). Missing references and related prior work flagged by reviewers should be added.

Rebuttal & Discussion: Reviewers initially questioned novelty and evaluation scope. The authors added experiments and new results, which addressed most concerns. These additional results and missing references should be integrated into the main text.

Decision: Accept (poster). The paper identifies a new failure mode, provides a practical solution, and demonstrates strong results.